# Layered Pd oxide on PdSn nanowires for boosting direct H₂O₂ synthesis

Hong-chao Li[1,2,5], Qiang Wan[3,5], Congcong Du[1,2], Jiafei Zhao[1,2], Fumin Li[1,2], Ying Zhang[1,2], Yanping Zheng[1], Mingshu Chen[1], Kelvin H. L. Zhang[1,2], Jianyu Huang[4], Gang Fu[1,2], Sen Lin[3] ✉, Xiaoqing Huang[1,2] ✉ & Haifeng Xiong[1,2] ✉

Hydrogen peroxide (H₂O₂) has the wide range of applications in industry and living life. However, the development of the efficient heterogeneous catalyst in the direct H₂O₂ synthesis (DHS) from H₂ and O₂ remains a formidable challenge because of the low H₂O₂ producibility. Herein, we develop a two-step approach to prepare PdSn nanowire catalysts, which comprises Pd oxide layered on PdSn nanowires (Pd$_L$/PdSn-NW). The Pd$_L$/PdSn-NW displays superior reactivity in the DHS at zero Celcius, presenting the H₂O₂ producibility of 528 mol kg$_{cat}^{-1}$·h$^{-1}$ and H₂O₂ selectivity of >95%. A layer of Pd oxide on the PdSn nanowire generates bi-coordinated Pd, leading to the different adsorption behaviors of O₂, H₂ and H₂O₂ on the Pd$_L$/PdSn-NW. Furthermore, the weak adsorption of H₂O₂ on the Pd$_L$/PdSn-NW contributes to the low activation energy and high H₂O₂ producibility. This surface engineering approach, depositing metal layer on metal nanowires, provides a new insight in the rational designing of efficient catalyst for DHS.

Hydrogen peroxide (H₂O₂) is widely used as oxidant in many fields because it is environmentally friendly and the byproducts only involve H₂O and O₂, as compared to other oxidants (e.g. Cl-containing oxidants and nitric acid)[1,2]. Currently, the anthraquinone oxidation process (AO process) is employed as the method to produce H₂O₂ in industry, whereas the AO process requires highly intense energy consumption and the addition of toxic organic solvents, such as alkylbenzene and so on. Therefore, it is desirable to develop sustainable process and more efficient strategy towards the production of H₂O₂[3–7].

Direct H₂O₂ synthesis (DHS) from H₂ and O₂ is an effective alternative for small-scale and on-site H₂O₂ production, which is usually performed using Pd-based catalyst[8–11]. However, the catalytic activity was limited due to the high-side reactions such as hydrogenation and decomposition when employing a pure Pd-based catalyst, leading to a decreased net yield. Numerous efforts have been devoted to developing high-performance catalysts towards DHS, such as the use of bimetallic Pd-Au[12–15], trimetallic Pd-Au-Pt[16], acid and halide additives[17–21]. However, these Pd catalysts contain expensive gold and the H₂O₂ yield is still far from that achieved by the AO process[22–24]. The addition of Sn to Pd catalysts has attracted intensive attention due to the high stability and inert hydrogenation with Sn component for the H₂O₂ synthesis[3]. In particular, the synthesis of PdSn nanocatalyst involved a multi-step protocol of oxidation-reduction-oxidation (O-R-O), and a tin oxide surface layer that encapsulates small Pd-rich particles was formed while leaving larger PdSn alloy particles exposed. The PdSn nanocatalyst prepared via O-R-O presented the H₂O₂ production with high selectivity of >95%, while only showing the H₂O₂ producibility of ~70 mmol·g$_{cat}^{-1}$·h$^{-1}$.

[1]The State Key Laboratory of Physical Chemistry of Solid Surfaces, iChEM (Collaborative Innovation Center of Chemistry for Energy Materials), Department of Chemistry, College of Chemistry & Chemical Engineering, Xiamen University, Xiamen 361005, China. [2]Innovation Laboratory for Sciences and Technologies of Energy Materials of Fujian Province (IKKEM), 4221 Xiangan South Road, Xiamen 361102, P. R. China. [3]State Key Laboratory of Photocatalysis on Energy and Environment, College of Chemistry, Fuzhou University, Fuzhou 350002, China. [4]Clean Nano Energy Center, State Key Laboratory of Metastable Materials Science and Technology, Yanshan University, Qinhuangdao 066004, China. [5]These authors contributed equally: Hong-chao Li, Qiang Wan. ✉e-mail: slin@fzu.edu.cn; hxq006@xmu.edu.cn; haifengxiong@xmu.edu.cn

Metal nanowires (NWs) is one-dimensional (1D) structure material, which has been utilized in photonic[25,26], electrical[27,28], and plasmonic-related applications[28,29]. As compared to nanoparticles or bulk material, these 1D nanowire structures can expedite orientable electronic or ions transfer and diffusion to promote catalytic kinetics. Therefore, nanowires can provide a unique platform to study catalysis[30,31]. For example, in a nanowire-bacteria hybrid system, nanowires can capture electrons and deliver them to bacteria, allowing bacteria to ensure the conversion of $CO_2$[26]. On the other hand, cobalt oxide (CoO) nanorods/nanowires were reported to create oxygen vacancies on the nanofacets[27]. The catalyst exhibits excellent electrocatalytic ORR/OER performance due to the modulated electronic structure of cobalt oxide nanorods/nanowires revealed by simulation.

Herein, we develop an approach to prepare PdSn nanowires to directly produce $H_2O_2$ and found that a layer of Pd oxide on a $Pd_4Sn$ alloy nanowire (NW) prepared via two-step synthesis (Fig. 1) present efficient reactivity in the direct production of $H_2O_2$. This approach involves the synthesis of the surface-rough $Pd_4Sn$ nanowires (PdSn-NW, Sn:Pd molar ratio is 4) by a solvothermal method firstly (Supplementary Fig. 1)[32]. Then, metal Pd precursor was deposited onto the $Pd_4Sn$ nanowires, followed by dispersing on the $TiO_2$ support. After rapid annealing of the material in air, an efficient catalyst for direct $H_2O_2$ synthesis was obtained and denoted as $Pd_L$/PdSn-NW catalyst (Supplementary Fig. 2). The unsupported $Pd_L$/PdSn-NW presents the morphology of worm-liked nanowires and some nanowires connect each other to form interconnected structures (Fig. 2a). The outmost surface of the structures contains Pd and Sn terminated with oxygen because both of the two metals were oxidized after the annealing in the air as shown in Fig. 1. We also synthesize and test the $TiO_2$-supported PdSn nanowire catalyst prepared via one-step synthesis (denoted as PdSn-NW). As can be seen, the two unsupported materials show similar nanowire morphology (Fig. 2a and Supplementary Fig. 1). After loading onto $TiO_2$, the nanowire morphologies do not change (see discussion below).

## Results and discussion

### Materials synthesis and catalytic performance

The catalytic reactivity of the $TiO_2$-supported PdSn nanowire catalyst synthesized by the two-step ($Pd_L$/PdSn-NW, 4.1 wt.%Pd loading) was evaluated in the direct synthesis of $H_2O_2$ at zero Celcius. In comparison, we also synthesized PdSn-NW catalyst via one-step (PdSn-NW), PdSn nanoparticle (PdSn-NP), and $SnO_x$ supported on PdSn-NW ($SnO_x$/PdSn-NW) via the two-step approach. The supports of all these reference catalysts are $TiO_2$ and all the activity data reported in the work are achieved from catalysts supported on $TiO_2$. Prior to dispersing on $TiO_2$, the morphologies of these reference nanowire/nanoparticles are well characterized using TEM (Supplementary Figs. 3, 4). It should be mentioned that both TEM-EDS and X-ray photoelectron spectroscopy (XPS) analysis indicate that there is no residual Br ion detected on the Pd catalyst (Supplementary Fig. 5) and therefore, the effect of the Br ion in the $H_2O_2$ synthesis can be excluded. The supported $Pd_L$/PdSn-NW catalyst synthesized by two-step shows the high $H_2O_2$ producibility of ~680 mol/kg h$^{-1}$ after annealing at 350 °C in the air (Fig. 2b). The addition of Sn onto the PdSn nanowires leads to the formation of $SnO_x$/PdSn-NW catalyst, which is less active than the supported $Pd_L$/PdSn-NW catalyst after annealing at 350 °C in air. Moreover, with the increase of Sn loading (4.5–13 wt.%), the reactivity of the $SnO_x$/PdSn-NW catalyst further decreases, indicating the addition of Sn on the PdSn nanowires has a negative effect on the reactivity of the PdSn-NW (Fig. 2b). However, the $Pd_L$/PdSn-NW catalyst annealing at 350 °C presents the high hydrogenation activity, which is undesirable for the DHS[3]. We investigated the effect of the annealing temperature on the reactivity of the $Pd_L$/PdSn-NW catalyst and found that annealing the $Pd_L$/PdSn-NW catalyst at 400 °C presented the $H_2O_2$ producibility of ~528 mol/kg h$^{-1}$ with the complete absence of hydrogenation (Fig. 2c). Therefore, 400 °C was employed as the annealing temperature for all supported catalysts in the following discussion, unless otherwise noted. Moreover, the Pd/Sn ratios and metal loadings of all the catalysts were measured by EDS and ICP-OES analysis, and a good agreement was obtained using the two techniques (Supplementary Figs. 6 and 7).

When using the two-step approach to deposit Pd precursor onto the as-synthesized PdSn nanowires to form $Pd_L$/PdSn-NW catalyst, the Pd loading is precisely controlled to cover the PdSn nanowires as monolayer (Supplementary Note 1) and two layers, which are denoted as $Pd_L$/PdSn-NW and $Pd_{2L}$/PdSn-NW, respectively. The annealed $Pd_L$/PdSn-NW catalyst shows superior reactivity than the PdSn-NW catalyst prepared via one-step (Fig. 2d and Supplementary Fig. 8). Although

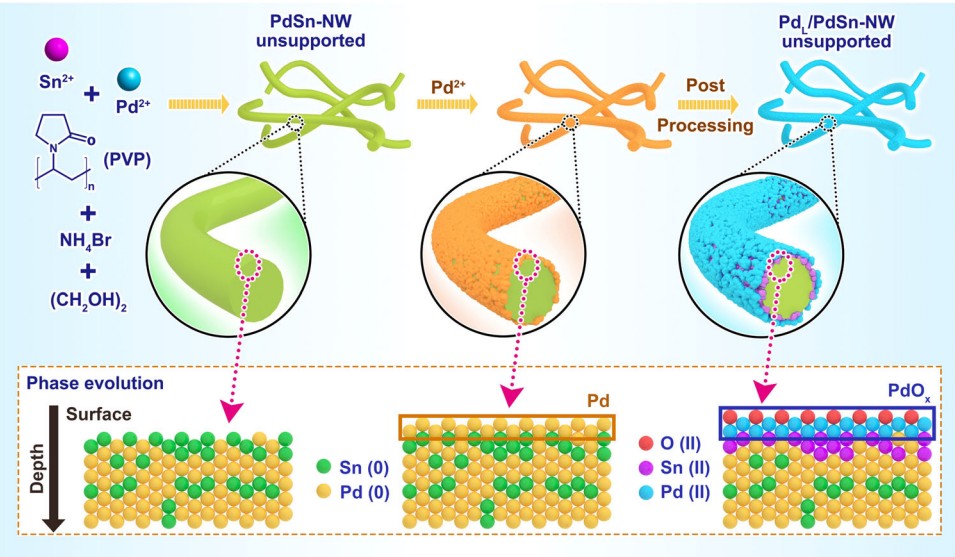

**Fig. 1 | Schematic illustration of the synthesis of the unsupported PdSn-NW and $Pd_L$/PdSn-NW.** The supported PdSn-NW and $Pd_L$/PdSn-NW catalysts are supported on $TiO_2$. The two-step protocol involving an annealing process was used for the synthesis of $Pd_L$/PdSn-NW. PdSn nanowire (NW) was prepared with a Pd: Sn molar ratio of 4 firstly and then, Pd precursor was deposited on the as-prepared PdSn nanowires again, followed by depositing on $TiO_2$ and a rapid annealing in air. A PdSn-NW catalyst with Pd:Sn ratio of 4 was also synthesized for the purpose of comparison via one-step method by mixing $Sn^{2+}$, $Pd^{2+}$, PVP, EG, and $NH_4Br$.

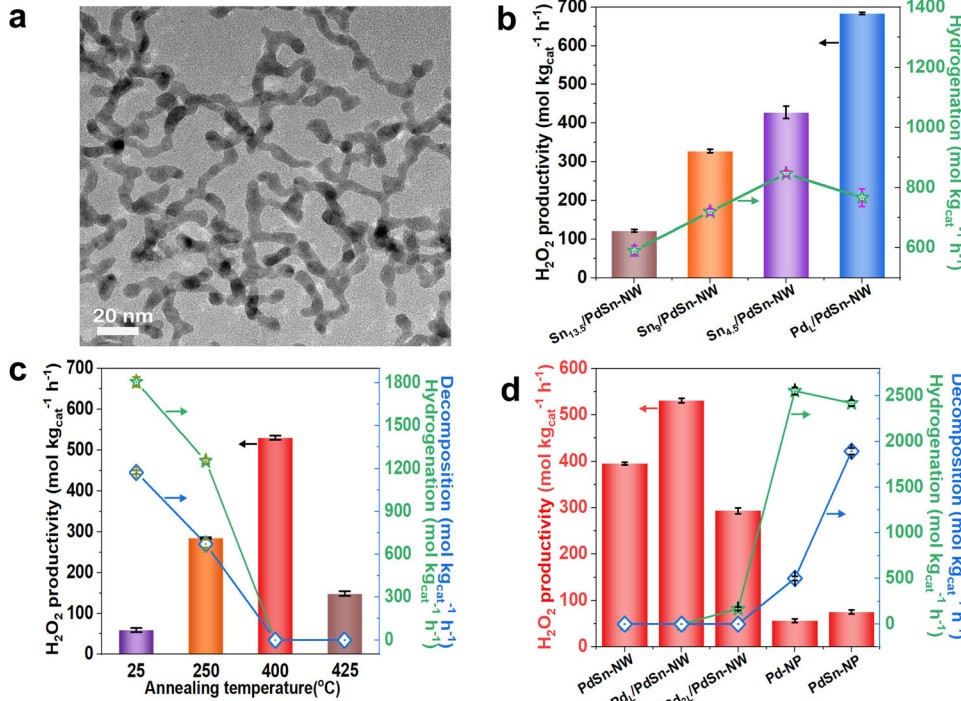

**Fig. 2 | Representative TEM image and the catalytic performances of the PdSn nanowire catalysts in the direct $H_2O_2$ synthesis (DHS). a** Representative TEM image of the unsupported PdSn nanowire catalyst prepared by two-step (unsupported $Pd_L$/PdSn-NW). **b** $H_2O_2$ producibility of the supported $Pd_L$/PdSn-NW and $Sn_x$/PdSn-NW catalysts with different Pd/Sn ratios after annealing in air (350 °C, 8 min), demonstrating the addition of Sn has negative effect on the producibility of $H_2O_2$. **c** $H_2O_2$ producibility, hydrogenation, and decomposition of the supported $Pd_L$/PdSn-NW catalyst annealing at different temperatures in air, showing that the

supported $Pd_L$/PdSn-NW annealing at 400 °C did not catalyze the hydrogenation and decomposition. **d** The comparison of the $H_2O_2$ producibility, hydrogenation, and decomposition of supported $Pd_L$/PdSn-NW catalyst with other Pd catalysts (Table 1). The error bars in **b**–**d** show the standard deviation in the measurements. The standard deviation was achieved from the repeated runs of three to five times using fresh catalyst each time. The error bars refers to the standard deviation of multiple times measurements of hydrogen peroxide productivity.

PdSn nanowire alone (PdSn-NW) shows no hydrogenation or decomposition activity (Fig. 2d), the $H_2O_2$ selectivity is only ~70% (Table 1). This is explained by the fact that the $H_2O_2$ selectivity is calculated from the first step in this two-step process (Supplementary Fig. 9). Furthermore, the annealed $Pd_L$/PdSn-NW catalyst presents outstanding catalytic activity for $H_2O_2$ production and higher $H_2O_2$ selectivity, as compared to $Pd_{2L}$/PdSn-NW catalyst (Fig. 2d). This indicates that the addition of the extra Pd onto the $Pd_L$/PdSn-NW has a negative effect on the $H_2O_2$ production. Besides, both the $H_2O_2$ hydrogenation and decomposition are completely inhibited on the annealed $Pd_L$/PdSn-NW catalyst, as compared to other active Pd catalysts reported. Furthermore, the annealed $Pd_L$/PdSn-NW catalyst is very stable in the

reaction and no deactivation is found in multiple runs (Supplementary Fig. 10). The reactivity of the supported $Pd_L$/PdSn-NW catalyst was also compared with other Pd-based catalysts (Fig. 2d and Table 1, Supplementary Tables 1 and 2), such as Pd nanoparticles, PdSn nanoparticles, and nice PdSn nanowires prepared by the approach reported in the literature[32]. None of them showed comparable $H_2O_2$ producibility and low decomposition/hydrogenation activity (Supplementary Fig. 11) as the supported $Pd_L$/PdSn-NW catalyst exhibiting the $H_2O_2$ selectivity of >95% under the same conditions. It should be mentioned that the $H_2O_2$ productivity slightly slows down over time (after 30 min) and it is partially ascribed to the decrease of the reactant gas concentration and the accumulation of $H_2O_2$ in the autoclave according to Le Chatelier's principle (Supplementary Fig. 12). Besides, it possibly suggests that the catalyst changes in the presence of high concentration $H_2O_2$ and either become less active or begins to degrade $H_2O_2$ at high $H_2O_2$ concentrations.

### Materials characterization

The composition and morphology of the $Pd_L$/PdSn-NW catalyst are first characterized by X-ray diffraction (XRD) and electron microscopy. XRD patterns of the unsupported PdSn nanowire prepared by two-step before and after annealing show the diffraction peaks located at lower diffraction angles (i.e., larger d spacings) as compared to that of metal Pd (Fig. 3a), indicating the crystal expansion of metal Pd due to the alloying of Sn atom. Annealing this unsupported PdSn nanowire at 350 °C or 400 °C does not make significant changes on the XRD patterns, indicating that the nanowire is thermally stable. When dispersing the nanowire on $TiO_2$, XRD pattern of the supported $Pd_L$/PdSn-NW catalysts only presents the diffraction peaks of the support $TiO_2$ (Fig. 3a and Supplementary Fig. 13), indicating the high dispersion of

**Table 1 | Catalytic performances of various Pd catalysts supported on $TiO_2$ after pretreating under different conditions in the direct synthesis of $H_2O_2$ from $H_2$ and $O_2$**

| Entry | Catalyst[a] | Annealing[b] | $H_2O_2$ Prod. (mol/kg h⁻¹) | $H_2O_2$ Sel. (%) | $H_2$ Conv.(%) |
|---|---|---|---|---|---|
| 1 | PdSn-NW | 400 °C, 8 min | 389 | 70.6 | 22.1 |
| 2 | $Pd_L$/PdSn-NW | 400 °C, 8 min | 528 | 95.3 | 22.1 |
| 3 | $Pd_{2L}$/PdSn-NW | 400 °C, 8 min | 290 | 38.3 | 30.5 |
| 4 | PdSn-NP | 400 °C, 8 min | 99 | 15.6 | 23.8 |
| 5 | PdSn-NP | 400 °C, 4 h | 22 | 6.9 | 12.1 |
| 6 | PdSn-NP | n.a. | 72 | 5.8 | 47.9 |
| 7 | Pd-NP | 400 °C, 8 min | 68 | 43.5 | 4.9 |
| 8 | Pd-NP | n.a. | 56 | 11.1 | 16.0 |

[a]All the catalysts are supported on $TiO_2$.
[b]n.a.: no annealing.

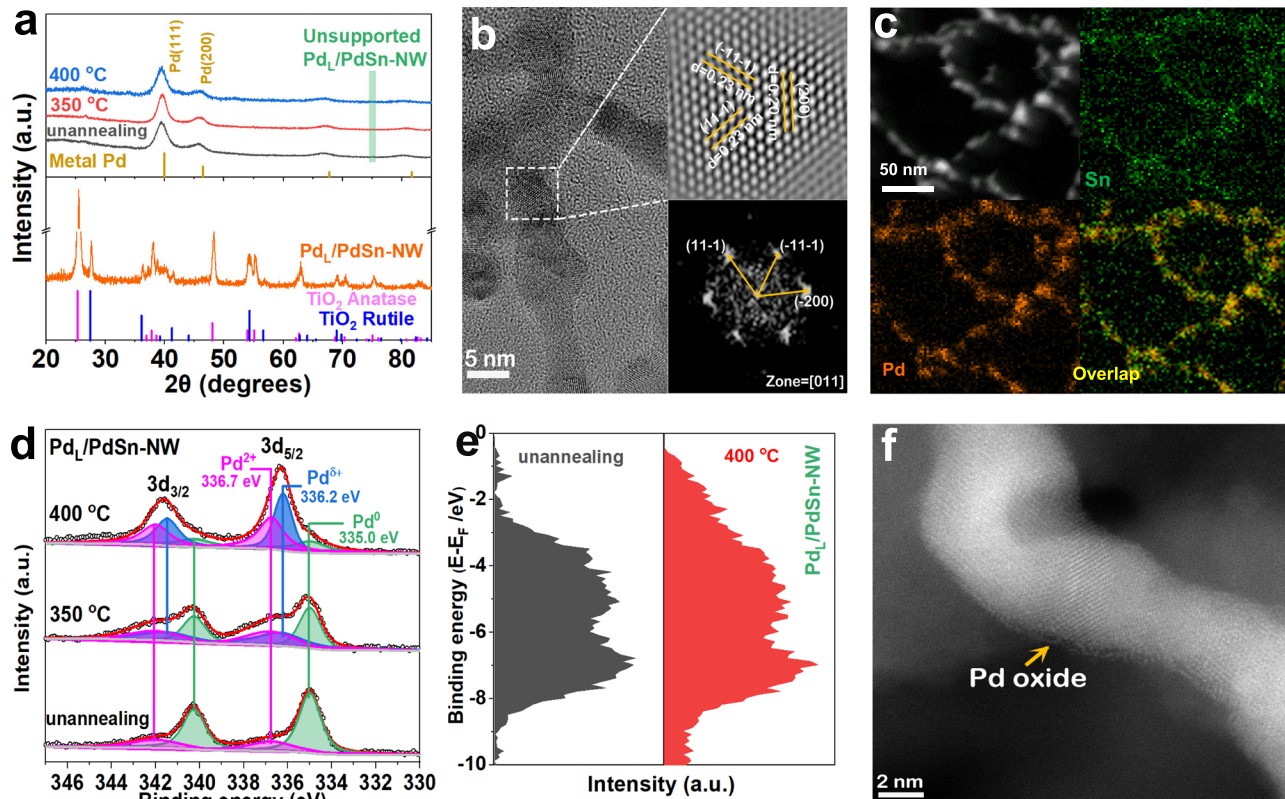

**Fig. 3 | Structural characterization of the unsupported and supported Pd_L/PdSn-NW catalysts before and after annealing in air at 400 °C. a** XRD patterns of the unsupported and supported Pd_L/PdSn-NW catalysts before and after annealing in air. **b** HRTEM image of the unsupported Pd_L/PdSn-NW sample showing the lattice fringes of the PdSn phase. **c** STEM image and STEM-EDS mapping of the supported Pd_L/PdSn-NW showing the close proximity of both Pd and Sn elements, indicating the successful synthesis of Pd_4Sn alloy. **d** XPS spectra of Pd 3*d* core level of the supported Pd_L/PdSn-NW catalyst before and after annealing in air. **e** Surface valence band photoemission spectra of the supported Pd_L/PdSn-NW catalysts annealing at different temperatures in air. **f** HAADF-STEM image of the supported Pd_L/PdSn-NW catalyst after annealing showing a layer of Pd oxide on the PdSn nanowires.

PdSn nanowire on the support. After annealing the supported Pd_L/PdSn-NW catalyst at different temperatures (350 °C and 400 °C, Supplementary Fig. 13a) in air, we do not find the presence of other diffraction peak or phase separation, indicating that the PdSn-NW and Pd_L/PdSn-NW catalysts are quite stable after the pretreatment. The representative high-resolution transmission electron microscopy (TEM)/scanning transmission electron microscopy (STEM) images of the Pd_L/PdSn-NW are shown in Fig. 3b, c. The nanowires have the diameter ranging from 5 to 10 nm and STEM-EDS line scanning shows the presence of both Pd and Sn in the nanowire (Supplementary Fig. 14). The high-resolution TEM of the nanowire is shown in Fig. 3b and the image exhibits the lattice fringes of 0.23 and 0.20 nm, corresponding to the (111) and (200) plane of Pd_4Sn phase, respectively (more images are shown in Supplementary Fig. 15). STEM-EDS mapping demonstrated that both Pd and Sn signals are uniformly dispersed in the nanowires (Fig. 3c). Therefore, the electron microscopy characterization of the annealed Pd_L/PdSn-NW sample demonstrates that the nanowire morphology of PdSn-NW is well maintained after loading onto TiO_2 and annealing in air.

XPS is used to investigate the surface species of the supported Pd_L/PdSn-NW catalyst during the annealing process (Fig. 3d, e, Supplementary Figs. 16–19). As shown in Fig. 3d, the XPS spectrum of the unannealed Pd_L/PdSn-NW catalyst shows that the main Pd species is Pd metal and the presence of a very small amount of Pd²⁺ prior to annealing. After annealing at 350 °C in air, the Pd²⁺ species content increases and Pd metal is still the dominant species on the surface. After annealing the Pd_L/PdSn-NW at 400 °C in air, we observe that the main Pd on the catalyst is Pd²⁺ species with a small amount of Pd metal, demonstrating the oxidation of metal Pd after annealing. Furthermore,

the XPS analysis indicates that the Sn species on the supported Pd_L/PdSn-NW catalyst is SnO_x (Supplementary Fig. 19) and the Pd/Sn ratio is ~2. Since Sn/PdSn-NW catalysts are less active and selective than PdSn-NW catalyst, the active site is not ascribed to the SnO_x. Therefore, the rapid annealing process generated Pd oxide on the PdSn nanowires, which is observed in the HAADF-STEM image (Fig. 3f and Supplementary Fig. 20). The d-band model has been widely used in order to understand activity trends in metal-surface-catalyzed reactions[33]. Here, the d-band center obtained from XPS was used to study the interaction between gas adsorption/intermediates and metal surface during annealing in air. The surface valence band photoemission spectrum of the annealed Pd_L/PdSn-NW is more delocalized and widely distributed than that of the unannealed Pd_L/PdSn-NW (Fig. 3e), indicating the different activation ability of gas molecules or intermediates on the annealed Pd_L/PdSn-NW catalyst[34]. The annealed Pd_L/PdSn-NW catalyst, therefore, provides Pd oxide to modulate the adsorption/desorption behaviors of the reactants on the catalyst surface, which was confirmed by the improved H_2O_2 producibility of Pd_L/PdSn-NW (Table 1) and density functional theory (DFT) calculation below.

The chemical environment of the Pd atoms in the supported Pd_L/PdSn-NW catalyst after annealing in the air was examined via X-ray adsorption spectroscopy (Fig. 4a and Supplementary Figs. 21 and 22). We also performed the XAS measurements of the Pd foil, PdO, PdSn-NW, and PdSn-NP catalysts for comparison. For the reference Pd foil, a major peak attributing to the Pd–Pd scattering was observed. The EXAFS spectra of PdSn-NW and PdSn-NP catalysts show a similar or slightly expanded scattering distance as compared to that of the metal Pd foil (Fig. 4a), which is attributed to the Pd-Pd and PdSn scattering. The Pd-K edge EXAFS of the supported Pd_L/PdSn-NW catalyst after

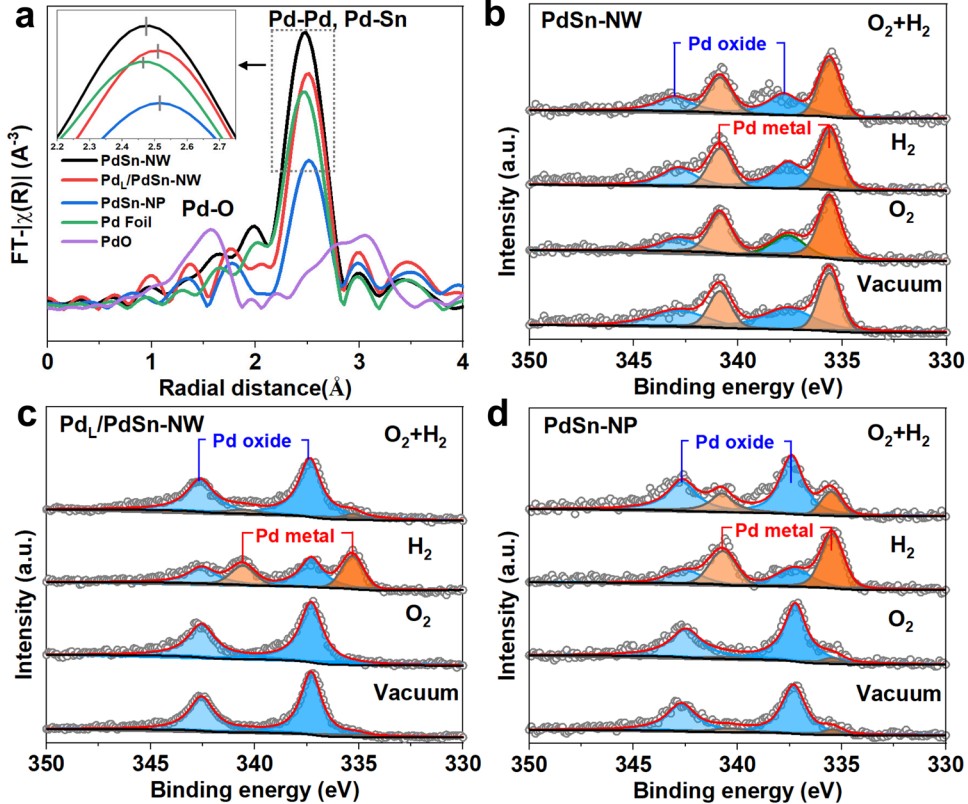

**Fig. 4 | EXAFS spectra of the PdK edge of PdSn catalysts and reference samples, and NAP-XPS of Pd 3d spectra for PdSn catalysts under different treatment conditions. a** EXAFS spectra of the PdK edge of PdSn catalysts after annealing in air. **b** In situ NAP-XPS of Pd 3d spectra for PdSn-NW in the presence of $O_2$, $H_2$, and

$O_2/H_2$. **c** In situ NAP-XPS of Pd 3d spectra for $Pd_L$/PdSn-NW after annealing in air in the presence of $O_2$, $H_2$ and $O_2/H_2$. **d** In situ NAP-XPS of Pd 3d spectra for PdSn-NP after annealing in air in the presence of $O_2$, $H_2$ and $O_2/H_2$.

annealing (air, 400 °C) also have a major peak corresponding to Pd-Pd and PdSn scattering (Fig. 4a), indicating the main Pd phase on the $Pd_L$/PdSn-NW catalyst is metal Pd and the rapid annealing process does not change the bulk phase of the PdSn nanowires. EXAFS fit gave a coordination number of 4.2 at a bond distance of 2.76 Å (Supplementary Table 3). The EXAFS is fitted to Pd-Pd and PdSn scattering features because the XRD results clearly indicate that the bulk phase of the PdSn nanowire after annealing in air is metal Pd (Fig. 3a) and both XRD and XAS are bulk techniques. Furthermore, the high-resolution TEM (HRTEM) image shows the lattice fringes corresponding to metallic PdSn nanowire (Supplementary Fig. 15a). Since XPS is more surface sensors for observing the changes under different conditions than XRD and XAS, We performed near-ambient pressure X-ray photoelectron spectroscopy (NAP-XPS) to in-situ investigate the surface properties of the PdSn catalysts (both PdSn nanoparticle and PdSn nanowire prepared via one-step and two-step approaches) in the flowing $O_2$, $H_2$ and $O_2/H_2$ mixture (Fig. 4b–d). Conventional PdSn nanoparticle catalyst showed the presence of Pd oxides in both vacumm and the flowing $O_2$, whereas metal Pd was formed after switching to $H_2$ or $H_2/O_2$ mixture (Fig. 4d), indicating that the conventional PdSn nanoparticles are not stable in $H_2$ and $H_2/O_2$ atmospheres. For the PdSn nanowire catalysts, the $Pd_L$/PdSn-NW prior to annealing mainly shows the metal Pd phase on the catalyst surface (Fig. 4b) and there is a very small change for the Pd 3d XPS spectra of the catalyst after flushing in $H_2$, $O_2$, and $H_2/O_2$, indicating that the unannealing $Pd_L$/PdSn-NW maintain the metal Pd surface in the flowing $H_2/O_2$. Likewise, the Pd 3d XPS spectrum of the $Pd_L$/PdSn-NW after annealing in air at 400 °C indicated that Pd oxide is the main Pd species on the surface at vacuum (Fig. 4c). The introduction of a flow of $O_2$ has no observed effect on the Pd 3d XPS spectrum of the annealed $Pd_L$/PdSn-NW. After introducing a flow of $H_2$, we observed the presence of

metal Pd, as well as the presence of Pd oxide. However, in the presence of $H_2/O_2$ mixture, the $Pd_L$/PdSn-NW surface after annealing was maintained as Pd oxide. Therefore, in-situ NAP-XPS evidenced that the surface species of the $Pd_L$/PdSn-NW after annealing is Pd oxide in the reaction feed of $H_2/O_2$, while the major Pd species on both PdSn-NP and PdSn-NW are metal Pd in the presence of $H_2/O_2$.

## Theoretical studies on reaction mechanism

Based on the above reactivity and characterization results, we propose that the excellent reactivity of the PdSn nanowires prepared by the two-step approach ($Pd_L$/PdSn-NW) is associated with the Pd oxide layer on PdSn nanowire which was detected via AC-STEM and NAP-XPS (Figs. 3 and 4). The stable Pd oxide layer on PdSn nanowire in the presence of $H_2/O_2$ exhibited enhanced reactivity as compared to the conventional PdSn nanoparticle and PdSn nanowire prepared by a one-step approach. To better understand the unique properties of the Pd oxide layer on the $Pd_L$/PdSn-NW in direct $H_2O_2$ synthesis, we performed DFT calculations based on three models of PdO(101), $Pd_4Sn$ and PdO monolayer supported on $Pd_4Sn(111)$ (PdO@$Pd_4Sn$) (Fig. 5a). The PdO(101) surface contains tetra-coordinated Pd (denoted as $Pd_{4c}$) and tri-coordinated Pd (denoted as $Pd_{3c}$) and the outmost surface atoms of the $Pd_4Sn$ model are Pd atom, while the PdO@$Pd_4Sn$ surface contains $Pd_{4c}$ and bi-coordinated Pd (denoted as $Pd_{2c}$). Details for model construction can be found in the computational method section. Bader charge analysis shows that $Pd_{4c}$ on PdO@$Pd_4Sn$ and PdO(101) are charged similarly with +0.83|e| and +0.85|e| (Supplementary Fig. 23), respectively. Nevertheless, the charges of $Pd_{2c}$ on PdO@$Pd_4Sn$ and $Pd_{3c}$ on PdO(101) are different, with the former being charged +0.46|e| and the latter having a charge of +0.67|e|. The surface Pd atoms in $Pd_4Sn$ are slightly positive by +0.11|e|, which is therefore more metallic than those in PdO(101) and PdO@$Pd_4Sn$. Thus, these

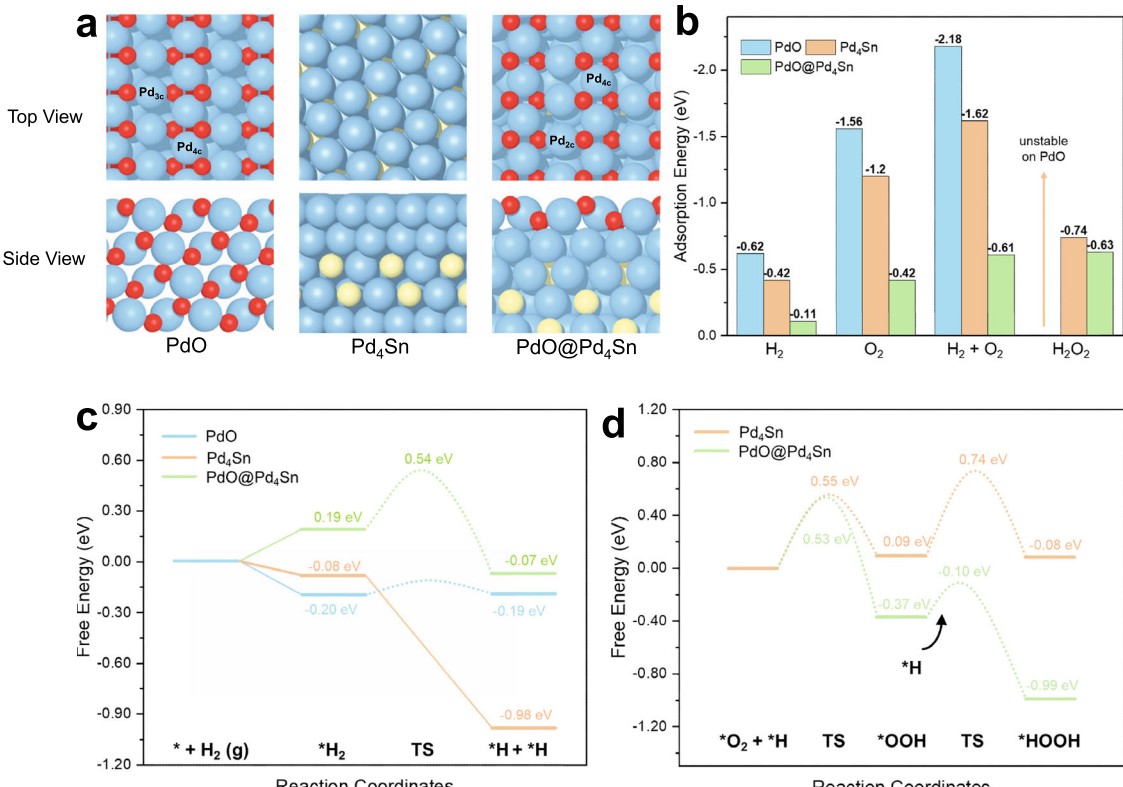

**Fig. 5 | Adsorption energy of key species and proposed mechanism. a** DFT optimized structures of PdO(101), Pd₄Sn, and PdO@Pd₄Sn with **b** adsorption energies of H₂, O₂, H₂ + O₂, and H₂O₂, **c** free energy profiles for H₂ activation, and **d** free energy profiles for O₂ reduction by the surface hydrides on these three models. TS transition state. Color code: Pd, blue; O, red; H, white; Sn, yellow.

three catalysts with different Pd chemical states may lead to different interactions between the adsorbate and the catalyst.

Subsequently, we investigated the adsorption behavior of the reaction molecules (O₂, H2, and H₂O₂) on PdO(101), Pd₄Sn, and PdO@Pd₄Sn. The adsorption configurations and corresponding adsorption energies are shown in Supplementary Fig. 24 and Fig. 5b, respectively. The calculation results show that all three molecules exhibit strong adsorption on the PdO(101) surface. H₂ prefers to adsorb on top of Pd₃c and O₂ tends to adsorb on the Pd₃c-Pd₃c bridge site, leading to adsorption energies of −0.62 and −1.56 eV, respectively. For H₂O₂, we found that it has a spontaneous dissociation to form two OHs on the two Pd₃c sites on PdO(101), suggesting that PdO(101) has poor selectivity for H₂O₂ synthesis. For PdO@Pd₄Sn, its binding to the three molecules is significantly weaker than that on PdO(101). H₂ is physically adsorbed on the Pd₂c top sites (2.76 Å away from Pd₂c) with an adsorption energy of −0.11 eV, the adsorption of O₂ on the Pd₂c-Pd₂c bridge sites leads to small adsorption energy of −0.42 eV. H₂O₂ adsorbs on Pd₂c-Pd₂c bridge sites, yielding moderate adsorption energy of −0.63 eV. Likewise, as for the Pd₄Sn surface, H₂ is adsorbed on the top of Pd while the O₂ molecule connects with four surface Pd atoms, having the adsorption strengths as those located on between PdO(101) and PdO@Pd₄Sn. To gain in-depth insight into the relation between electronic structure and catalytic behavior, the projected density of states (PDOSs) of the surface Pd- $d_{z^2}$ states on PdO(101) and PdO@Pd₄Sn were calculated and compared since the $d_{z^2}$ states are considered to play an important role in molecule adsorption[35–37]. As can be seen in Supplementary Fig. 25, the distribution of $d_{z^2}$ of surface Pd₃c on PdO is more localized than that of surface Pd₂c on PdO@Pd₄Sn. Furthermore, a sharp peak of $d_{z^2}$ states exists across the Fermi level on PdO(101), implying strong interaction between surface Pd and the reaction species via donating/accepting electrons. In contrast, on PdO@Pd₄Sn, the distribution of Pd-$d_{z^2}$ state is less significant around

the Fermi level, indicating the weaker binding of reaction species to Pd. It is therefore proposed that different adsorption behaviors of O₂, H₂, and H₂O₂ on the three models lead to different catalytic performances in the direct synthesis of H₂O₂.

In the following, we investigated the reaction mechanism of the O₂ reduction on PdO(101), Pd₄Sn, and PdO@Pd₄Sn. First, we calculated the activation of the most stable adsorption state of H₂, whose reaction pathways on these three models are shown in Fig. 5c. On PdO(101), H₂ dissociation occurs readily with an energy barrier of 0.09 eV. In the final state, the two dissociated H atoms are adsorbed on the Pd₃c-Pd₃c bridge site and the Pd₃c top site, respectively. In addition, it is found that the migration of H between two adjacent Pd₃c sites is facile with a small energy barrier of 0.11 eV, while the migration of H from the Pd₃c site to an adjacent oxygen site is difficult, requiring an energy barrier of 0.87 eV(Supplementary Fig. 26). It indicates that Pd-H hydride species are stable on PdO(101). On Pd₄Sn, H₂ activation on Pd₄Sn is barrierless, similar to the previous DFT findings on Pd(111) surface[38] (Supplementary Fig. 27). On PdO@Pd₄Sn, the cleavage of H-H bond in the most stable H₂ adsorbed has to overcome an energy barrier of 0.35 eV. Similar to the case on PdO(101), the Pd-H species is also stable because the energy barrier for H migration between Pd₂c sites (0.69 eV) is lower than that (1.07 eV) of its migration to oxygen sites (Supplementary Fig. 28). Therefore, H₂ readily dissociates on all of PdO(101), Pd₄Sn, and PdO@Pd₄Sn to produce stable Pd-H hydride, which provides a hydrogen source for the subsequent O₂ reduction (Fig. 5d). We also calculated other possible H₂ dissociation pathways but they possess higher energy barriers (Supplementary Figs. 29 and 30). Then, we investigate the O₂ reduction to H₂O₂ (energy profile in Fig. 5d and configurations in Supplementary Figs. 31 and 32). On the Pd₄Sn, the energy barriers of the first (*O₂ + *H → *OOH) and second (*OOH + * H → *H₂O₂) hydrogenation of O₂ is 0.55 eV and 0.65 eV with the reaction energies of 0.09 eV and −0.01 eV, respectively. On PdO@Pd₄Sn,

the $*O_2 + *H \rightarrow *OOH$ reaction encounters an energy barrier of 0.53 eV, while the subsequent $*OOH + *H \rightarrow *H_2O_2$ process has a lower energy barrier of 0.27 eV. This result indicates that the reaction of $O_2$ reduction to produce $H_2O_2$ has a lower rate-determining energy barrier on $PdO@Pd_4Sn$ than that on $Pd_4Sn$, in agreement with the experimental observations that $PdO@Pd_4Sn$ has better catalytic performance than $Pd_4Sn$ prepared by one-step. We did not infer the reaction pathway on PdO(101) because $H_2O_2$ was proved to be unstable on this surface (Fig. 5b and Supplementary Fig. 25), leading to low $H_2O_2$ production.

In addition, we simulated three main side reactions of $*O_2$, $*OOH$, and $*H_2O_2$ dissociation on $Pd_4Sn$ and $PdO@PdSn$, and the corresponding structures of IS, TS, and FS are shown in Supplementary Figs. 33 and 34. On $Pd_4Sn(111)$, the $*O_2$ and $*OOH$ dissociations are barrierless while the $*HOOH$ dissociation has a small barrier of 0.07 eV (Supplementary Fig. 33), suggesting the poor selectivity of $H_2O_2$ on $Pd_4Sn(111)$. On $PdO@Pd_4Sn$, the energy barriers of the three side reactions are 1.74, 0.53, and 0.32 eV (Supplementary Fig. 34), respectively, which are larger than those of the corresponding competitive reactions (0.53, 0.27, and 0.14 eV for $O_2$ hydrogenation, OOH hydrogenation, and $H_2O_2$ desorption, respectively, Supplementary Figs. 32, 34). Therefore, these results suggest the high selectivity of $H_2O_2$ on $PdO@Pd_4Sn$, which is consistent with the experimental observations (Fig. 2d and Table 1).

Selective production of $H_2O_2$ is a challenge in the direct $H_2O_2$ synthesis (DHS) from $H_2$ and $O_2$. In this contribution, we develop a two-step approach to prepare PdSn nanowires to efficiently catalyze direct $H_2O_2$ synthesis (DHS). This approach involves the first synthesis of the surface-rough $Pd_4Sn$ nanowire (PdSn-NW) by a solvothermal method. Then Pd precursor was deposited onto the PdSn alloy nanowire (NW), followed by annealing in air. The as-prepared $Pd_L$/PdSn-NW via the two-step approach presents efficient reactivity with $H_2O_2$ producibility of >520 mol $kg_{cat}^{-1}$ $h^{-1}$ and selectivity of >95% in the direct production of $H_2O_2$ at zero Celcius. For the PdSn nanowire prepared by one-step method and PdSn nanoparticle catalysts, the catalyst surfaces are prone to be reduced in the flowing $H_2/O_2$ and show low $H_2O_2$ reactivity.

The excellent $H_2O_2$ production over the $Pd_L$/PdSn-NW catalyst is attributed to the presence of Pd oxide layer on the PdSn nanowires. The Pd oxide layer is stable against reduction or oxidation in the flowing $H_2/O_2$, while other PdSn nanoparticles and PdSn nanowire catalysts undergo a reduction in the presence of $H_2/O_2$. The layered Pd oxide enables less adsorption of oxygen/hydrogen and decreases the rupture of both O-O and H-H bonds, as well as less adsorption of peroxide produced, leading to the complete inhibition of the $H_2O_2$ hydrogenation and decomposition. Therefore, engineering the surface of PdSn nanowires via two-step approach can generate a layer of Pd oxide on the PdSn nanowire, presenting excellent reactivity in the direct $H_2O_2$ synthesis, which provides a promising strategy to design and develop highly active DHS catalyst.

## Methods

### Chemicals

Bis (acetylacetonato) palladium (II) ($Pd(acac)_2$, 99%), tin(II) acetate ($Sn(Ac)_2$, 95%), methanol ($CH_3OH$, HPLC grade), titanium dioxide ($TiO_2$, P25, 99%) were purchased from Sigma-Aldrich. Palladium(II) nitrate dihydrate ($Pd(NO_3)_2 \cdot 2H_2O$, 99%), polyvinylpyrrolidone (PVP, MW = 58000), Tin(IV) chloride pentahydrate ($SnCl_4 \cdot 5H_2O$, 99%) were purchased from J&K Scientific Ltd. Ethylene glycol (EG, analytical grade), N,N-Dimethylacetamide (DMAC, analytical grade), ammonium bromide ($NH_4Br$, analytical grade) were purchase from Sinopharm Chemical Reagent Co. Ltd.(Shanghai, China). The de-ionized water (DI $H_2O$,18 MΩ/cm) used in all experiments was obtained by passing through an ultra-pure purification system. All the chemicals were used without further purification.

### Synthesis of unsupported PdSn nanowires via one-step

In a typical synthesis, the PdSn NWs ($Pd_4Sn$, marked as PdSn) were synthesized according to our previous report[32]. Briefly, 7.6 mg of $Pd(acac)_2$, 1.5 mg of $Sn(Ac)_2$, 15 mg of $NH_4Br$, 100 mg of PVP, 2 mL of DMAC, and 8 mL of EG were added into a vial (30 mL). And then, the mixture was sonicated for 15 min to ensure form a homogeneous solution. The solution was heated and maintained at 180 °C for 2 h in an oil bath. After that, the mixture was cooled down to room temperature. Finally, the precipitate was centrifuged and washed with ethanol/acetone mixture.

### Synthesis of unsupported $Pd_x$/PdSn-NW and $Sn_y$/PdSn-NW via two-step

In a typical synthesis, the $Pd_x$/PdSn or $Sn_y$/PdSn NWs were synthesized by the following steps. Briefly, 7.6 mg of $Pd(acac)_2$, 1.5 mg of $Sn(Ac)_2$, 15 mg of $NH_4Br$, 100 mg of PVP, 2 mL of DMAC, and 8 mL of EG were added into a vial (30 mL). Then, the mixture was sonicated for 15 min to ensure form a homogeneous solution. The solution was heated and maintained at 180 °C for 2 h in an oil bath. After that, the mixture was cooled down to room temperature to achieve the unsupported PdSn nanowires. The role of $NH_4Br$ in the synthesis of these worm structures is to induce the formation of the PdSn nanowires. Subsequently, the desired amount of $Pd(acac)_2$ or $Sn(Ac)_2$ solution (including 320 μL of EG and 80 μL of DMAC, e.g. 1.9 mg $Pd(acac)_2$) was added into the above solution. Next, the mixture was continuously stirred and held for 30 min at room temperature, followed by heating to 150 °C (hold for 2 h) with vigorous stirring. Then, the mixture was cooled down to ambient temperature again. Finally, the precipitate was centrifuged and washed with ethanol/acetone mixture. The precipitate achieved with the addition of 1.9 mg Pd precursor in the second step is denoted as unsupported $Pd_L$/PdSn-NW and the sample achieved with the addition of $Sn(Ac)_2$ solution in the second step is denoted as unsupported $SnO_x$/PdSn-NW.

### Synthesis of unsupported Pd nanoparticles

The unsupported Pd NPs were synthesized from a procedure similar to that of PdSn nanowires synthesized via one-step, but without the addition of $Sn(Ac)_2$.

### Synthesis of unsupported PdSn nanoparticles (PdSn-NP)

The bimetallic PdSn NP catalysts were synthesized according to a previous report[3]. In brief, 62.5 mg of $Pd(NO_3)_2 \cdot 2H_2O$, 2 mL of DI water were added into a vial (30 mL). The solution was heated to 80 °C with continuous stirring. Then, 1 mL of DI water solution (including 73.8 mg of $SnCl_4 \cdot 5H_2O$) was added to the vial and held for 15 min under stirring. Next, 0.6 g of $TiO_2$ and 1 mL of DI water were added to the solution, and the mixture was heated to 110 °C with continuous stirring in the air to evaporate the water. Subsequently, the solid sample was placed in a tube furnace and heated to 500 °C for 3 h under static air (heating rate of 10 °C/min). After cooling down, the solid sample was further heated to 200 °C for 2 h in a flow of 5% $H_2/Ar$ (heating rate of 10 °C/min). Then, the solid sample was annealed in static air at 400 °C for 8 min or 4 h, respectively (heating rate of 10 °C/min). The resultant was denoted as PdSn-NP.

### Synthesis of catalysts supported on $TiO_2$

Typically, the commercial $TiO_2$ material (anatase) and 6 mL of $CH_3Cl$ were added into a 30 mL vial. The PdSn-NW or nanoparticle materials were firstly dispersed in 6 mL of $CH_3Cl$ and ultrasonicated for about 5 min. The resulting mixture was added to the $TiO_2$ vial. Next, the mixture was continuously ultrasonicated for 30 min. Finally, the products were collected by centrifugation and dried at 60 °C overnight.

## Annealing treatment of the supported catalysts

The as-synthesized catalysts were treated via a rapid annealing process. Firstly, the tube furnace was heated to 400 °C and held for 30 min under static air. Next, the appropriate amount of catalyst was placed in the porcelain boat and pushed into the tube furnace. The catalyst left for several minutes by annealing and was taken out (8 min) for cooling. The final products were denoted as $Pd_L$/PdSn-NW (supported on $TiO_2$, the "NW" represents nanowires) and collected for further characterization.

## Catalyst characterization

The crystallographic structure of all the samples was determined by Powder XRD (Rigaku Ultima IV, Japan) patterns using a Cu Kα X-ray source ($\lambda = 1.54056$ Å). The morphology of all the samples was imaged by transmission electron microscope (TEM, JEM-1400, JEOL Co., Japan). The HRTEM, line-scan analysis, and elemental mapping were carried out on FEI Tecnai F30 electron microscope with an acceleration voltage of 200 kV. STEM images were obtained on an FEI Titan Themis Z, operating at 60-300 kV to observe the surface morphology of the materials. Before microscopy examination, the catalyst powders were ultrasonically dispersed in ethanol and then a drop of the solution was put onto a carbon-coated copper grid. The elemental content of all the samples was examined by scanning electron microscopy energy dispersive spectrometer (SEM-EDS, ZEISS Sigma, Germany) and inductively coupled plasma-atomic emission spectrometry (ICP-AES, iCap 7000, USA). The chemical compositions and valence states of all the samples were analyzed by XPS (K-Alpha, USA). NAP-XPS (SPECS Surface Nano Analysis GmbH, Germany) measurements were carried out on a SPECS system equipped with a differentially pumped Phoibos hemispherical electron energy analyzer using monochromatic Al Kα radiation (1486.6 eV). The $H_2$ or $O_2$ (99.999%) flow was introduced into the NAP cell and the total pressure was kept constant at 0.2 mbar via the electronic back-pressure regulator. X-ray Absorption Fine Structure (XAFS) of all samples was analyzed by TLS-01C beamline of the National Synchrotron Radiation Research Center (NSRRC, Hsinchu, Taiwan), and data were processed according to standard procedures using the Demeter program package (Version 0.9.24)[39].

## Direct $H_2O_2$ synthesis

All the catalysts supported on $TiO_2$ were tested in the direct $H_2O_2$ synthesis. Direct $H_2O_2$ synthesis experiment was carried out in a stainless-steel semibatch autoclave with a volume of 50 mL. Briefly, 5 mg of supported catalyst, 5.54 g of MeOH, and 3.0 g of $H_2O$ (both HPLC grade) were added to the autoclave. Then purged three times with $O_2$ (0.4 MPa) to the autoclave, and with filled with $O_2$ (0.4 MPa) and 5% $H_2$/Ar (3.6 MPa) until the total pressure of 4.0 MPa at room temperature. All experiments were performed in an ice-water bath and kept continuous stirring (1200 rpm) of reaction time for 15 min. The reaction was carried out at zero Celsius because it is much safer and easier to carry out in the lab. We also investigated the reactivity of the catalyst at room temperature for comparison. The producibility of the $Pd_L$/PdSn-NW catalyst at room temperature is slightly lower than that performed at zero Celcius in the direct $H_2O_2$ synthesis (Supplementary Fig. 35). Control experiments are also performed by varying the stirring rate from 100-1200 rpm, and varying the catalyst mass from 3 to 10 mg using autoclaves having different volumes (50 mL and 100 mL). All these experiments show similar $H_2O_2$ producibility under these conditions. Therefore, it indicates that there are no interphase mass transfer constraints in the reaction under the present conditions (Supplementary Fig. 36).

## Quantification of gas product

The gas products were detected via a gas chromatograph (GC, 8890, Agilent) equipped with a thermal conductivity detector (TCD) and a MolSieve 5 A packing column (G3591-80022). Typically, after the direct

$H_2O_2$ synthesis process, the autoclave was directly connected into the GC. The gas products were purged into GC by Ar.

The $H_2$ conversion was calculated as follows:

$$H_2 \text{conversion} = \frac{n(H_2)_{in} - n(H_2)_{out}}{n(H_2)_{in}} \times 100\% \quad (1)$$

## $H_2O_2$ productivity evaluation

The $H_2O_2$ productivity was analyzed by titrating with acidified $Ce(SO_4)_2$ (0.05 M) and using ferroin (about 100 μL) as indicator[3,40]. The acidified $Ce(SO_4)_2$ solutions were standardized against $(NH_4)_2Fe(SO_4)_2 \cdot 6H_2O$ using ferroin as an indicator. The $H_2O_2$ productivity and $H_2O_2$ selectivity were calculated based on the equations below:

$$H_2O_2 + 2Ce^{4+} \rightarrow 2Ce^{3+} + 2H^+ + O_2$$

$$H_2O_2 \text{selectivity} = \frac{n(H_2O_2)_{out}}{n(H_2O_2)_{out} + n(H_2O)_{out}} \times 100\% \quad (2)$$

$$\text{Where } n(H_2O)_{out} = n(H_2)_{converted} - n(H_2O_2)_{out} \quad (3)$$

## $H_2O_2$ hydrogenation and decomposition

$H_2O_2$ hydrogenation and decomposition were carried out using similar procedures to direct $H_2O_2$ synthesis. Typically, for $H_2O_2$ hydrogenation test ($H_2O_2 + H_2 \rightarrow 2H_2O$), 5 mg of supported catalyst (support is $TiO_2$), 5.54 g of MeOH, 2.0 g of $H_2O$ and 1.1 g $H_2O_2$ (30%) were added in the autoclave. Then, 3.6 MPa of $H_2$/Ar (5% $H_2$) was filled into the autoclave. Afterward, the reaction time extended until 30 min. For the $H_2O_2$ decomposition test ($H_2O_2 \rightarrow H_2O + O_2$), 5 mg of catalyst, 5.54 g of MeOH, 2.0 g of $H_2O$, and 1.1 g $H_2O_2$ (30%) were added to the autoclave. Then, 3.6 MPa of $N_2$ was filled into the autoclave. Subsequently, the reaction was carried out for 30 min. The $H_2O_2$ hydrogenation or decomposition rate was calculated based as follows:

$$H_2O_2 \text{hydrogenation/decomposition} = \frac{n(H_2O_2)_{in} - n(H_2O_2)_{out}}{W_{catalyst} \times t} \times 100\% \quad (4)$$

## Computational details

All spin-polarized DFT calculations were carried out by using the Vienna Ab initio Simulation Package. Interactions between ion cores and valence electrons were described by the projected augmented wave method and the exchange-correlation function was by the Perdew–Burke–Ernzerhof functional based on generalized gradient approximation[41]. Wavefunctions were expanded by the plane wave basis with a cutoff energy of 400 eV[42]. The van der Waals interactions were included via DFT-D2 in Grimme's scheme[43]. Transition states were determined by the Climbing Image Nudged Elastic Band and Dimer method[44,45], and charge states were obtained from Bader Charge analysis[46]. For all models (crystallographic information files of $Pd_4Sn$, PdO@$Pd_4Sn$, and PdO(101) were provided in supplementary materials as Supplementary Data 1–3, respectively), Monkhorst–Pack mesh with $2 \times 2 \times 1$ grid was adopted to sample the Brillouin zone[47]. Convergence criteria for structural optimizations were set to $10^{-5}$ eV and 0.02 eV/Å for energy and force, respectively. And for transition states searching, the criterion for force was set to 0.05 eV/Å. The $(2 \times 3)$-PdO(101) surface was cleaved from the optimized PdO unitcell ($a = b = 3.056$ Å, $c = 5.381$ Å, in agreement with experimental values ($a = b = 3.043$ Å and $c = 5.335$ Å)), containing 4 O-Pd-O tri-atomic layers. The bottom two layers were fixed while the other layers with surface species were fully relaxed in structural optimization. The lengths of $a$ and $b$ of supercell are 12.38 and 9.17 Å, respectively. Since the Pd unit is composed of 4 Pd

atoms, uniform $Pd_xSn_y$ can only be constructed as $Pd_2Sn_2$ and $Pd_3Sn$. In addition, our experiments show that nanowires expose more Pd on the surface, we, therefore, simulate the slab with 4 layers of uniform $Pd_3Sn$ covered with one layer of Pd (Supplementary Fig. 37). Then, the total Pd/Sn ratio equals 4:1. For all slabs, a vacuum space of 15 Å is adopted along the $z$-direction to avoid the interaction between periodic images.

The Gibbs free energy ($G$) is obtained by VASPKIT code[48] via the following formula:

$$G = E + E_{ZPE} + \Delta U_{0 \to T} - T \times S \qquad (5)$$

in which $E$ is the electronic energy of DFT calculations, $E_{ZPE}$ and $S$ are the zero-point energy and entropy computed from vibrational frequency analysis. T was the temperature adopted in our experiment, 273.15 K. $\Delta U_{0 \to T}$ is the internal energy difference between 0 and T K.

## Data availability

All data needed to evaluate the conclusions in the paper are present in the paper and/or the Supplementary Information. The raw data sets used for the presented analysis within the current study are available from the corresponding authors on reasonable request. Source data are provided with this paper.

## Code availability

The code that supports the findings of this study is available from the corresponding author upon reasonable request.

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

## Acknowledgements

Funds from the National High-Level Young Talents Program, National Natural Science Foundation of China (22072118 and 21973013), and National Natural Science Foundation of Fujian Province, China (2020J02025) are acknowledged. We also thank the financial support from State Key Laboratory for Physical Chemistry of Solid Surfaces and National Engineering Laboratory for Green Chemical Productions of Alcohols-Ethers-Esters of Xiamen University. Part fund was supported by Science and Technology Projects of Innovation Laboratory for Sciences and Technologies of Energy Materials of Fujian Province (IKKEM) (HRTP-[2022]-3) and the Fundamental Research Funds for the Central Universities (20720220008). S.L. thanks "Chuying Program" for the Top Young Talents of Fujian Province. J.H. would like to thank National Natural Science Foundation of China (Nos. 51772262, U20A20336, 21935009) and Natural Science Foundation of Hebei Province (No. B2020203037). Numerical computations were performed at Hefei advanced computing center. The authors thank Shanghai Synchrotron Radiation Facility (Shanghai Institute of Applied Physics) and beamline TLS-01C (Taiwan National Synchrotron Radiation Research Center) for providing the beam time.

## Author contributions

X.H., S.L., and H.X. conceived and supervised the research; H.-C.L. designed and performed the experiments; Q.W. performed and S.L. conducted the DFT calculations; C.D. and J.H. performed the HRTEM measurements; J.Z. performed the XAS analysis; Y.Zha. and F.L. help for the materials synthesis; M.C., Y.Zhe., K.Z., and G.F. helped with the data interpretations; H.-C.L., S.L., and H.X. wrote and edited the paper. All authors discussed the results and commented on the paper.

## Competing interests

The authors declare no competing interests.
