## [Peer Review File · Nature Communications]

Title: Layered Pd oxide on PdSn Nanowires for Boosting Direct H₂O₂ SynthesisREVIEWER COMMENTS

Reviewer #1 (Remarks to the Author):

The paper reports the synthesis of a PdSn nanowire structure with a PdO over layer which shows impressive performance for H₂O₂ direct synthesis with high selectivity. Beyond the headline catalytic result there are some ambiguities on the levels of structural analysis which lead to the hypothesis proposed for the high activity in addition to limited catalytic testing. For this reason, I do not recommend publication at this point but to consider major revisions based on the comments below.

- It is unclear in the paper if the samples are supported on TiO₂ or not – the STEM images show the unsupported Pd/PdSn materials – but the experimental mentions supported the samples on TiO₂ - what is the morphology of the supported samples if in fact these are the ones tested?
- Following this it is unclear how much Pd is used in each reaction – the experimental states 5 mg of catalyst – but not if this is the un-supported samples in which case the Pd content is high or a loading for the supported samples.
- The paper reports tests at 15 mins reaction time – as the H₂O₂ synthesis reaction scheme is a sequential reaction scheme longer times should be investigated to prove high selectivity in the presence of accumulated H₂O₂. Additionally, the comparison against literature in the SI should only take into account reactions at the same time point. As many systems can be shown to be selective at short times.
- In addition re-use tests would strengthen the catalytic testing section and show that the materials are in fact structurally stable.
- Following this – all graphs are labelled yield of h₂O₂ when in fact the report productivity or rate – the authors should look to include the actual amount of h₂O₂ (as ppm or wt%) to show if this is significant.
- The synthesis method contains a significant amount of Br – a known selectivity promoter for H₂O₂ synthesis over Pd – do the authors know the levels of residual halide?
- The results in figure 1d suggest that the PdSn nanowire alone is selective – with no hydrogenation or decomposition activity which goes against the hypothesis that an over layer of PdO is required for high selectivity. The results in Table 1 seem to contradict this and show a H₂ selectivity of only 70%.
- This raises my main concern with the paper – the lack of characterisation of the Sn species. The analysis such as XPS is solely focused on the Pd species – and if the nanowires are less than 10 nm XPS can be assumed to be a bulk technique limiting information about the pure surface composition. Can the authors exclude the formation of amorphous SnO_x species similar to those reported by Hutchings – either under annealing at 400C or on exposure to aqueous environments? Pd:Sn ratios should be extracted – Auger analysis can help to identify the Sn species here.
- Scheme 1 seems to show a allow of oxidised Sn and Pd terminated with O. Is this a realistic structure?

- The XRD shows lattice expansion compared to pure Pd on formation of a proposed Pd₄Sn alloy however the EXAFS shows a similar or even contracted Pd-Pd scattering distance – how can this be reconciled? Would additional scattering features from Sn also be expected to be seen?

- EXAFS fitting – the EXAFS fitting reported in the SI needs to be revisited – in some cases the Debye Waller factors – a measure of disorder – are negative which is not possible – meaning these fits are not realistic and forcing the Pd-Sn distances/co-ordination is may not be suitable or a good model to use.

Reviewer #2 (Remarks to the Author):

Li et al. report on the preparation of alloyed, worm-like Pd/Sn nanostructures which were coated by PdO or SnO_x in a two-step procedure. The morphology of the wires was worm-like, defect-rich and rather irregular. Various analytical tools were used for characterization including TEM (HRTEM, STEM-HAADF with elemental mapping), SEM-EDS, ICP-OES, XRD, XPS, and XAS, in addition to theoretical calculations. The nanostructures were deposited on TiO₂ and used as catalysts in the direct synthesis of H₂O₂ from molecular H₂ and O₂.

There several points the authors should address to improve the investigation:

- 1) There are some mistakes concerning English spelling and grammar, which should be revised prior to publication.
- 2) Line 52, page 2: "...In particular, the synthesis of the PtSn nanocatalyst..." This refers probably rather to the PdSn catalyst and should be corrected.
- 3) Line 90, page 3: "...In comparison, we also synthesized PdSn-NW catalyst via one-step (PdSn-NW), PdSn nanoparticle (PdSn-NP) and SnO_x supported on PdSn-NW (SnO_x/PdSn-NW) via the two-step approach (Supplementary Figure 2-3). ..." It is not clear how these abbreviations relate to the samples shown in Figures 2 and 3 of the SI. Do these TEM images show the structure of the calcined samples?
- 4) Scheme 1: What is the role of NH₄Br in the synthesis of these worm structures? Ethylene glycol should also be mentioned here since it seems to be a prerequisite to form the structures.
- 5) Table 1 in the SI: The authors nicely compare their catalytic results with previous work of others in this area. Since the overall catalytic performance is highly dependent on the actual reaction conditions, they should further include more details of the reaction conditions (such as pressures, temperature etc.). The metal loading is reported just for some selected examples in this table and not for their own catalysts.

Methods:

- 6) Line 392, page 23: Why should the autoclave used herein be a semibatch autoclave?
- 7) Line 411, page 14: Reaction arrow should be used here.
- 8) Line 412, page 14: How did the authors determine the H₂O concentration needed to calculate the H₂O₂ selectivity according to the equation reported in line 412?

9) The standard deviations of the measurements are impressively small (Figure 1). Does the standard deviation here refer to several repetitions of the same catalytic test? How often did the authors repeat each catalytic test?

10) What is the metal loading of the TiO₂-supported catalysts? Figure 5 in the SI: "...Loadings of different Pd samples measured by ICP-OES analysis. ..." To which of the Pd samples do the authors refer here (i.e. the TiO₂-supported or unsupported worm structures)? Please, specify.

11) Did the authors perform control experiments to examine the possibility of interphase mass transfer constraints? If not, please state that explicitly.

I was wondering if the authors also tested the very nice nanowires that they synthesized by a similar procedure in Ref. 32 and if a similar effect of structural defects is also observed for direct H₂O₂ synthesis?

While this work shows some interesting results, significant revisions are required before publication in Nature Communications.

Reviewer #3 (Remarks to the Author):

Li et al. reports PdSn nanowire catalysts for efficient H₂O₂ direct synthesis where the nanowire is layered with Pd oxides. The nanowire catalysts provide superior reactivity in the H₂O₂ direct synthesis at zero Celcius, in particular, H₂O₂ selectivity of >95%. They claim that the superior catalytic performance mainly results from the Pd oxides layered on PdSn nanowires. Although the topic is timely and interesting, I have several critical concerns which prevent the publication of this work. My comments are found below.

1. The role of the Pd oxides for H₂O₂ direct synthesis is clear. In particular, in DFT results (Figure 4), I don't make sure that the PdSn nanowires with Pd oxide layers provides the high H₂O₂ selectivity of >95%. In Figure 4d, the authors considered only the main reaction for H₂O₂ direct synthesis without considering side reactions (*OH formation, *O₂ dissociation, and dissociation of the formed H₂O₂). Without comparing the side reaction pathways, one cannot make sure the high H₂O₂ selectivity. And, in Figure 4d, the *HOOH formation on PdO@Pd₄Sn is thermodynamically exothermic by ~1.25 eV, which indicates that the detachment of the formed H₂O₂ is likely a rate-determining step for H₂O₂ direct synthesis on PdO@Pd₄Sn. In this work, the catalytic performance was evaluated at zero Celcius. Thus, I don't think that the 1.25 eV can be overcome at zero Celcius, which revealing a disagreement between the DFT calculations and experiments.

2. Although the authors performed XPS analyses and Bader charge calculations, the origin of Pd oxide layers for the superior catalytic performance is not still clear. The authors need to perform in-depth electronic structure analyses with DFT methods and then compare with XPS analyses.

3. In the Introduction section, the authors explained that nanowire structure would be able to provide a

unique platform for catalysis. Because of this reason, the authors considered PdSn nanowire structures. However, in this work, the superior catalytic performance mainly results from Pd oxides layered on PdSn nanowires, not the nanowire structures themselves.

4. The authors evaluated catalytic performance of the nanowires for H₂O₂ direct synthesis at zero Celcius. Why did they consider zero Celcius? Usually, in the field of the catalysis for H₂O₂ direct synthesis, a room temperature has been considered. The authors need to evaluate catalytic properties of the samples at room temperatures and then compare with reported other state-of-the-art catalysts.

5. The authors need to show the data regarding stability of the nanowire catalyst such as recyclability or/and long-term stability.

<Minor comments>

1. In DFT calculations, why did consider the PdO(101) or Pd₄Sn(111) surfaces? Are the surfaces thermodynamically most stable? And, the authors need to clarify the crystal structures of PdO and Pd₄Sn for DFT calculations.

2. On line 52, the "PtSn" nanocatalyst should be changed into "PdSn".

Response to the Reviewers' comments

GENERAL: We thank the reviewers for all their comments, which greatly help us to improve the manuscript. We have considered all of them carefully and have revised the manuscript accordingly. The changes are colored in red in the manuscript. The detailed point-to-point responses are listed below.

Reviewer #1 (Remarks to the Author):

The paper reports the synthesis of a PdSn nanowire structure with a PdO over layer which shows impressive performance for H₂O₂ direct synthesis with high selectivity. Beyond the headline catalytic result there are some ambiguities on the levels of structural analysis which lead to the hypothesis proposed for the high activity in addition to limited catalytic testing. For this reason, I do not recommend publication at this point but to consider major revisions based on the comments below.

- It is unclear in the paper if the samples are supported on TiO₂ or not – the STEM images show the un-supported Pd/PdSn materials – but the experimental mentions supported the samples on TiO₂
- what is the morphology of the supported samples if in fact these are the ones tested?

Reply: We apologize for the confusion. In the original submission, the Fig. 1a and Fig. 2b in the manuscript and the Supplementary Figures 1, 2 and 3a are unsupported PdSn materials. The Fig. 2c and Fig. 2f in the manuscript and the Supplementary Figures 8, 9 and 13 are the PdSn nanowire catalyst supported on TiO₂. All the reactivity data presented in this manuscript are achieved from the supported catalysts. In this revised submission, we have added more statements to make it clear in the main manuscript and Supplementary materials.

Revisions:

In the line 95 of the revised manuscript, we have added “The supports of all these reference catalysts are TiO₂ and all the activity data reported in the work are achieved from catalysts supported on TiO₂.”

In the revised Supplementary Materials, we have added more TEM images below to further show the morphology of the supported sample.

Supplementary Figures for the reviewers TEM images of the supported Pd_L/PdSn-NW catalyst (the support is TiO₂).

- Following this it is unclear how much Pd is used in each reaction – the experimental states 5 mg of catalyst – but not if this is the un-supported samples in which case the Pd content is high or a loading for the supported samples.

Reply: We apologize for the confusion. Again, only supported catalysts were tested in the direct synthesis of H₂O₂ and all the activity data are achieved from the catalysts supported on TiO₂ in this work. Therefore, 5 mg SUPPORTED catalyst was used in each reaction. We have added the Pd content in the revised Supplementary Materials and added more texts to make this issue clear in the revised manuscript.

Revisions:

We have added the Pd loading into the Table 1 in the revised Supplementary Materials.

- The paper reports tests at 15 mins reaction time – as the H₂O₂ synthesis reaction scheme is a sequential reaction scheme longer times should be investigated to prove high selectivity in the presence of accumulated H₂O₂. Additionally, the comparison against literature in the SI should only take into account reactions at the same time point. As many systems can be shown to be selective at short times.

Reply: We thank the reviewer for the suggestions. We have therefore tested the catalytic performances of the catalyst for longer times and the results are shown in the figure below. The H₂O₂ selectivity of the Pd_I/PdSn-NW catalyst for the longer times (≥ 30 min) are $> 90\%$. These results are included in the revised Supplementary materials. In addition, according to the reviewer's comment, more information was added in Table S1 and the comparison against literature in the SI is compared at the same time point now.

Supplementary Figure for the reviewers The function of the H₂O₂ producibility with the time over the Pd_I/PdSn-NW catalyst in direct H₂O₂ synthesis from H₂ and O₂ at zero Celsius.

Supplementary Table 1. Comparison of catalytic reactivity of PdSn nanowire catalyst in this work and the well-developed catalysts in literature for the direct H₂O₂ synthesis

Catalyst	Pd loading (wt.%) ^c	Synthesis Conditions	H ₂ O ₂ Prod. (mol/kg _{cat} *h) ^d	H ₂ O ₂ Prod. (mol/kg _{pd} *h)	Hydrog/Decomp Conditions	H ₂ O ₂ hydrogenation (mol/kg _{cat} *h)	H ₂ O ₂ decomposition (mol/kg _{cat} *h)	Ref.
Pd _I /PdSn-NW	4.1 (0.9)	4.0 MPa (5% H ₂ , 10% O ₂ , 85% Ar), 0 °C, 5 mg cat., 50 mL autoclave	528 (373)	12878	3.6 MPa (5% H ₂ , 95% Ar) / 3.6 MPa N ₂ , 0 °C, 5 mg cat.	0	0	This work
Pd _I /PdSn-NW ^a	4.1 (0.9)	4 MPa (5% H ₂ , 10% O ₂ , 85% Ar), 0 °C, 5 mg cat., 60 mL autoclave	(170.1)	5667	3.6 MPa (5% H ₂ , 95% Ar) / 3.6 MPa N ₂ , 0 °C, 5 mg cat.	~260	~0.9	ACS Catalysis 2021, 11 , 1106
Pd ₆ Pb NRs/TiO ₂	3.1 (1.1)	4.0 MPa (5% H ₂ , 10% O ₂ , 45% Ar, 40% He), 0 ± 2 °C, 30 mg cat., 30 mL autoclave	(320)	12800	No test / 2 MPa (20% O ₂ , 80% He), 0 ± 2 °C, 30 mg cat.	/	41	ACS Catalysis 2021, 11 , 1946
Pd ₃ Pb/s-TiO ₂	2.4 (1.7)	40 bar (4% H ₂ , 20% O ₂ , 76% N ₂), 30 °C, ca. 32 mg cat., 300 mL autoclave	(176.1)	7339	/	/	/	ACS Catalysis 2021, 11 , 2288
1% Pd NP/C	1.3 (/)	40 bar (5% H ₂ , 25% O ₂ , 70% CO ₂), 2 °C, 10 mg cat., 100 mL autoclave	(120)	12000	29 bar (5% H ₂ , 95% CO ₂) / No test, 2 °C, 10 mg cat.	360	/	ACS Catalysis 2020, 10 , 5928

R-PdNi/TiO ₂ -C ^b	0.85 (0.42)	1 atm (2.5% H ₂ , 50% O ₂ , 47.5% Ar), 20 °C, 5 mg cat., 100 mL reactor	(96.6)	11364	/	/	/	ACS Catalysis 2021, 11 , 8407
3 wt % Pd-2 wt % Sn/TiO ₂	3 (2)	4.0 MPa (5% H ₂ , 25% O ₂ , 70% CO ₂), 2 °C, 10 mg cat., 100 mL autoclave	(61)	2033	29 MPa (5% H ₂ , 95% CO ₂)/1 atm (air), 2 °C, 10 mg cat.	0	/	Science 2016, 351 , 965
2.5% Au-2.5% Pd/carbon	2.5 (2.5)		(160)	6400		0	0	Science 2009, 323 , 1037

^a Annealing temperature is 350 °C

^b The catalytic reaction solvent contains 50 mL D₂O and 0.5 M sulfuric acid

^c The parenthesis is the loading of the second metal in the Pd-M alloy catalysts

^d The parenthesis is the producibility of the catalysts performed in 30 min.

Revisions:

We have changed Table 1 in the Supplementary materials to the above Table.

We have added the above figure into the revised Supplementary Materials.

- In addition re-use tests would strengthen the catalytic testing section and show that the materials are in fact structurally stable.

Reply: According to the reviewer's suggestion, we have performed the re-use tests for the supported Pd_L/PdSn-NW catalyst. The results are shown below and displayed in the revised Supplementary information now. It is seen that the supported Pd_L/PdSn-NW catalyst shows stable activity in multiple runs, indicated the nanowires are structurally stable.

Supplementary Figure for the reviewers The recyclability of the Pd_L/PdSn-NW catalyst in the direct synthesis of H₂O₂. C/C₀ is the ratio of H₂O₂ producibility produced in each run to that of the first run in the reaction cycles. The spent catalyst after each run was centrifuged and washed using ethanol/acetone mixture. After drying at 60 °C, the material was used for the next-cycle test. During the recycle experiments, there is the catalyst loss in each run and the catalyst mass of each cycle in the experiments of recycling is shown in the table below:

Catalyst	Catalyst mass of each cycle (mg)				
	1	2	3	4	5

Pd _L /PdSn-NW	5.07	4.84	4.58	4.27	3.99
PdSn-NP	5.02	4.87	4.64	4.32	4.09

Revisions:

In the line 126 of page 4 of the revised manuscript, we have added “Moreover, the Pd_L/PdSn-NW catalyst is stable under the reaction conditions and there is no distinct deactivation found through the multiple recycle experiments (Supplementary Figure 9).”

In the revised Supplementary materials, we have added the recyclability figure and the Table shown above.

- Following this – all graphs are labelled yield of H₂O₂ when in fact the report productivity or rate – the authors should look to include the actual amount of H₂O₂ (as ppm or wt%) to show if this is significant.

Reply: According to the reviewer’s suggestion, we have changed the labelled yield of H₂O₂ to productivity in the revised manuscript. We have also analyzed the actual amount of H₂O₂ as ppm and wt.% to investigate if this is significant. As shown in the Table below, the improvement in the producibility of H₂O₂ over the supported Pd_L/PdSn-NW catalyst is significant when including the actual amount of H₂O₂ as ppm or wt.%, as compared to other Pd catalysts.

Supplementary Table for the reviewers The actual amount of H₂O₂ producibility of the Pd catalysts as ppm and wt.% in direct H₂O₂ synthesis from H₂ and O₂ at zero Celsius.

Catalyst	H ₂ O ₂ productivity (mol/kg*h)	Reaction time (min)	Concentration (ppm)	Concentration (wt.%)
Pd _L /PdSn-NW	528	15	2244	0.26
Pd _L /PdSn-NW	373	30	3171	0.37
PdSn-NW	398	15	1692	0.20
PdSn-NP	99	15	420	0.05
Pd-NP	68	15	289	0.03
3 wt.%Pd-2 wt.%Sn/TiO ₂ *	61	30	1037	0.21

*: *Science* 2016, **351**, 965

Revisions:

We have added the above Table in the revised Supplementary Materials.

In the line 128 of the revised manuscript, we have changed “Supplementary Table 1” to “Supplementary Tables 1 and 2”.

- The synthesis method contains a significant amount of Br – a known selectivity promoter for H₂O₂ synthesis over Pd – do the authors know the levels of residual halide?

Reply: Yes, the synthesis method contains a significant amount of Br. However, the materials were washed for multiple times using solvent to remove the Br ion. We have checked the levels of

residual halide by TEM-EDS and XPS analysis and the amount of residual Br ion is out of the detection limit as shown below.

Supplementary Figure for the reviewers EDS (a) and XPS (b) analysis of the supported Pd_L/PdSn-NW catalyst showing the absence of Br residual.

Revisions:

To address the reviewer's comment, we have added "It should be mentioned that both EDS and XPS analysis indicates that there is no residual Br ion detected on the Pd catalyst (Supplementary Figure 5) and therefore, the effect of the Br ion in the H₂O₂ synthesis can be excluded." in line 94 of the revised manuscript.

In the revised Supplementary materials, we have also added the above EDS and XPS spectra.

- The results in figure 1d suggest that the PdSn nanowire alone is selective – with no hydrogenation or decomposition activity which goes against the hypothesis that an over layer of PdO is required for high selectivity. The results in Table 1 seem to contradict this and show a H₂O₂ selectivity of only 70%.

Reply: We apologize for the confusion. Yes, the PdSn nanowire alone (PdSn-NW in the manuscript) has no hydrogenation or decomposition activity (Figure 1d), while the selectivity of H₂O₂ for this nanowire alone is as low as 70.6% in the direct H₂O₂ synthesis (Table 1). This issue raised is because of the unclear clarification in the original submission. As shown in the Scheme below, the H₂O₂ synthesis reaction includes two processes: the direct H₂O₂ synthesis and the following dehydrogenation/decomposition. Our results showed that the PdSn nanowire catalyst alone did not hydrogenate or decompose H₂O₂ in the second step, while the selectivity of H₂O₂ in the direct H₂O₂ synthesis from H₂ and O₂ is 70.6% and the other product is H₂O (29.4%) in the first step.

Supplementary Scheme for the reviewers Reaction routes in the direct H_2O_2 synthesis reaction.

Revisions:

To address the reviewer's comment, we have added the above Scheme into the revised Supplementary materials.

In the line 118 of the revised manuscript, we have added "Although PdSn nanowire alone (PdSn-NW) shows no hydrogenation or decomposition activity (Fig. 1d), the H_2O_2 selectivity is only ~70% (Table 1). This is explained by the fact that the H_2O_2 selectivity is calculated from the first step in this two-step process (Supplementary Scheme 1)."

- This raises my main concern with the paper – the lack of characterisation of the Sn species. The analysis such as XPS is solely focused on the Pd species – and if the nanowires are less than 10 nm XPS can be assumed to be a bulk technique limiting information about the pure surface composition. Can the authors exclude the formation of amorphous SnO_x species similar to those reported by Hutchings – either under annealing at 400C or on exposure to aqueous environments? Pd:Sn ratios should be extracted – Auger analysis can help to identify the Sn species here.

Reply: We thank the reviewer for the suggestion. According to the reviewer's suggestion, we have performed the Auger analysis of the supported $\text{Pd}_L/\text{PdSn-NW}$ catalyst and the spectra collected are displayed below. As can be seen, the signal/noise ratio of the Sn spectrum is very poor, and the Pd/Sn ratio for the supported $\text{Pd}_L/\text{PdSn-NW}$ catalyst is ~1.2.

Supplementary Figure for the reviewers Auger spectra of Pd and Sn of the supported $\text{Pd}_L/\text{Pd-NW}$ catalyst after annealing in air.

According to the reviewer's comment, we have also performed the Sn XPS analysis and the Sn 3d spectra of the supported Pd_L/PdSn-NW catalyst before and after annealing are shown below. As can be seen, the surface of the supported Pd_L/PdSn-NW catalyst before annealing contains both metal Sn and Sn oxide. The presence of Sn oxide is due to the oxidation of metal Sn during the ex-situ XPS measurement. After annealing in air, the majority of the Sn species on the supported Pd_L/PdSn-NW catalyst is Sn oxide. The Pd/Sn ratio for the supported Pd_L/PdSn-NW catalyst after annealing is ~2. Therefore, SnO_x species are formed after annealing and the information achieved matches well with the model demonstrated in Scheme 1. However, as demonstrated in Fig. 1b, the presence of the SnO_x species on the nanowire is not the active site because the other SnO_x/PdSn-NW catalysts are less active and selective than Pd_L/PdSn-NW catalyst in the direct H₂O₂ synthesis.

Supplementary Figure for the reviewers XPS spectra of Sn 3d of the supported Pd_L/Pd-NW catalyst before and after annealing in air.

Revisions:

To address the reviewer's comment, we have added "Furthermore, the XPS analysis indicates that the Sn species on the supported Pd_L/PdSn-NW catalyst is SnO_x (Supplementary Fig. 17) and the surface Pd/Sn ratio is ~2. Since SnO_x/PdSn-NW catalysts are less active and selective than PdSn-NW catalyst, the active site is not ascribed to the SnO_x." in the line 184 of the revised manuscript.

We have also added the Sn 3d XPS spectra (figure above) into the revised Supplementary materials.

- Scheme 1 seems to show a allow of oxidised Sn and Pd terminated with O. Is this a realistic structure?

Reply: Yes, our characterization results infer that both Pd and Sn species on the catalyst surface are metal oxide after the rapid annealing. In this work, a two-step process was used to prepare the efficient Pd_L/PdSn-NW catalyst, involving depositing Pd on PdSn nanowires firstly, followed by a rapid annealing in air. Because metal Pd is oxidized in air, we infer that the outmost layer of the material is Pd oxide. As demonstrated by XPS, the majority of Sn was also oxides after the

annealing process, therefore, the structure in Scheme 1 shows the oxidized Sn and Pd terminated with O.

Revisions:

To address the reviewer's concern, we have added "The outmost surface of the structures contains Pd and Sn terminated with oxygen because both of the two metals were oxidized after the annealing in air as shown in Scheme 1." in the line 74 of the revised manuscript.

- The XRD shows lattice expansion compared to pure Pd on formation of a proposed Pd₄Sn alloy however the EXAFS shows a similar or even contracted Pd-Pd scattering distance – how can this be reconciled? Would additional scattering features from Sn also be expected to be seen?

Reply: We agree with the reviewer that the XRD shows lattice expansion compared to pure Pd on formation of a proposed Pd₄Sn alloy. In Figure 3a, the EXAFS spectra of both supported PdSn-NW and Pd_L/PdSn-NW catalysts show similar or slightly expanded Pd-Pd scattering distance, as compared to that of Pd foil. We agree with the reviewer that the scattering features for the PdSn-NW and Pd_L/PdSn-NW catalysts are attributed to the scattering of Pd-Pd and Pd-Sn, instead of only Pd-Pd in the original submission. We have therefore changed the statements and made revisions on the assignment of the scattering peak in Figure 3a.

Revisions:

To address the reviewer's comment, we have added "We also performed the XAS measurements of the Pd foil, PdO, PdSn-NW and PdSn-NP catalysts for the comparison. For the reference Pd foil, a major peak attributing to the Pd-Pd scattering was observed. The EXAFS spectra of PdSn-NW and PdSn-NP catalysts show the similar or slightly expanded scattering distance as compared to that of the metal Pd foil (Fig. 3a), which is attributed to the Pd-Pd and Pd-Sn scattering." in the line 211 of the revised manuscript.

We have also changed "Pd-Pd" to "Pd-Pd, Pd-Sn" in the text of Figure 3a.

We have also added an insert figure and a dash line to demonstrate the scattering distances in the Figure 3a.

- EXAFS fitting – the EXAFS fitting reported in the SI needs to be revisited – in some cases the debye waller factors – a measure of disorder – are negative which is not possible – meaning these fits are not realistic and forcing the Pd-Sn distances/co-ordination is may not be suitable or a good model to use.

Reply: We thank the reviewer for pointing out this issue. We have therefore revisited the EXAFS fitting reported in the SI. A couple of EXAFS spectra with negative Debye-Waller factors have been refitted, and the updated data are shown in the revised Supplementary Table 3.

Revisions:

We have refitted the EXAFS spectra and the updated data are shown in the revised Supplementary Table 3, as shown below.

Supplementary Table for the reviewers Fit parameters of Pd K-edge EXAFS for PdSn catalysts and references.

Samples	Path	N ^a	R/Å ^b	$\sigma^2/\text{Å}^{-2}$ ^c	$\Delta E/\text{eV}$ ^d	R-factor ^e (%)
Pd _L /PdSn-NW	Pd-Pd	4.2 (0.3)	2.76 (0.01)	0.003 (0.001)	0.91 (0.5)	1.6
	Pd-Sn	0.8 (0.1)	2.57 (0.02)	0.003 (0.001)		
PdSn-NP	Pd-Pd	3.3 (0.2)	2.74 (0.01)	0.003 (0.002)	-2.45 (0.6)	2.3
	Pd-Sn	0.13 (0.1)	2.53 (0.02)	0.01 (0.001)		
Pd foil	Pd-Pd	12 (0.4)	2.74 (0.02)	0.005 (0.001)	-7.71 (1.4)	1.4
PdO	Pd-Pd	4 (0.4)	3.04 (0.03)	0.005 (0.001)	-4.31 (1.2)	1.5
	Pd-O	4 (0.3)	2.00 (0.01)	0.002 (0.001)	-4.31 (1.2)	

^a Coordination number

^b Interatomic distance

^c Debye-Waller factor

^d Shift of the energy threshold

^e R factor indicating the goodness of fit between experimental and theoretical data

Reviewer #2 (Remarks to the Author):

Li et al. report on the preparation of alloyed, worm-like Pd/Sn nanostructures which were coated by PdO or SnO_x in a two-step procedure. The morphology of the wires was worm-like, defect-rich and rather irregular. Various analytical tools were used for characterization including TEM (HRTEM, STEM-HAADF with elemental mapping), SEM-EDS, ICP-OES, XRD, XPS, and XAS, in addition to theoretical calculations. The nanostructures were deposited on TiO₂ and used as catalysts in the direct synthesis of H₂O₂ from molecular H₂ and O₂.

There several points the authors should address to improve the investigation:

1) There are some mistakes concerning English spelling and grammar, which should be revised prior to publication.

Reply: We have looked through the whole manuscript and revised some mistakes concerning English spelling and grammar. We thank the reviewer for pointing out these issues.

2) Line 52, page 2: "...In particular, the synthesis of the PtSn nanocatalyst..." This refers probably rather to the PdSn catalyst and should be corrected.

Reply: We thank the reviewer for pointing out this typo and we have changed "PtSn" to "PdSn" in the line 52 of page 2 in the revised manuscript.

3) Line 90, page 3: "...In comparison, we also synthesized PdSn-NW catalyst via one-step (PdSn-NW), PdSn nanoparticle (PdSn-NP) and SnO_x supported on PdSn-NW (SnO_x/PdSn-NW) via the two-step approach (Supplementary Figure 2-3). ...". It is not clear how these abbreviations relate to the samples shown in Figures 2 and 3 of the SI. Do these TEM images show the structure of the calcined samples?

Reply: We apologize for the confusion on the abbreviations of the catalysts. In the original submission, the Fig. 1a and Fig. 2b in the manuscript and the Supplementary Figure 1, Figure 2 and Figure 3a are unsupported PdSn samples. The Fig. 2c and Fig. 2f in the manuscript are the PdSn

nanowire catalyst supported on TiO₂. All the reactivity data tested in this manuscript are supported catalysts. In the revised manuscript, we have added more details in the revised manuscript to clarify this issue.

Revisions:

We have made clear clarification about these abbreviations of the catalysts in the revised manuscript.

In the line 95 of the revised manuscript, we have added “The supports of all these reference catalysts are TiO₂. Prior to disperse on TiO₂, the morphology of these nanowire/nanoparticles were characterized using TEM (Supplementary Figure 3-4)”.

In the caption of the revised Supplementary Figure 2, we have added “These TEM images show the morphologies of the nanowires prior to loading on TiO₂ support.”

In the caption of the revised Supplementary Figure 3, we have added “These TEM images show the morphologies of the nanoparticles before and after loading on TiO₂ support.

4) Scheme 1: What is the role of NH₄Br in the synthesis of these worm structures? Ethylene glycol should also be mentioned here since it seems to be a prerequisite to form the structures.

Reply: NH₄Br has the effect as an ionic capping agent selectively adsorbing onto the facets of Pd nanocrystal to induce the formation of smaller nanorods/nanowires (*J. Am. Chem. Soc.* 2007, 129, 12, 3665; *Adv. Mater.* 2007, 19, 3385). Therefore, the role of NH₄Br in the synthesis of these worm structures is to induce the formation of the PdSn nanowires. Ethylene glycol is a reduction agent in the Scheme 1. According to the reviewer’s suggestion, we have also added ethylene glycol in the revised Scheme 1.

Revisions:

In the Method section, we have added “The role of NH₄Br in the synthesis of these worm structures is to induce the formation of the PdSn nanowires.”

We have added ethylene glycol in the revised Scheme 1.

5) Table 1 in the SI: The authors nicely compare their catalytic results with previous work of others in this area. Since the overall catalytic performance is highly dependent on the actual reaction conditions, they should further include more details of the reaction conditions (such as pressures, temperature etc.). The metal loading is reported just for some selected examples in this table and not for their own catalysts.

Reply: We agree with the reviewer that the overall catalytic performance is highly dependent on the actual reaction conditions. According to the reviewer’s comment, we have included more details of the reaction conditions (pressures and temperature) in the revised Table 1 in the SI (see below). We have also added the metal loading of our own catalysts in the revised Table 1 in the SI.

Supplementary Table 1. Comparison of catalytic reactivity of PdSn nanowire catalyst in this work and the well-developed catalysts in literature for the direct H₂O₂ synthesis

Catalyst	Pd loading (wt.%) ^c	Synthesis Conditions	H ₂ O ₂ Prod. (mol/kg _{cat} *h) ^d	H ₂ O ₂ Prod. (mol/kg _{pd} *h)	Hydrog/Decomp Conditions	H ₂ O ₂ hydrogenation (mol/kg _{cat} *h)	H ₂ O ₂ decomposition (mol/kg _{cat} *h)	Ref.
----------	--------------------------------	----------------------	---	---	--------------------------	--	--	------

Pd ₁ /PdSn-NW	4.1 (0.9)	4.0 MPa (5% H ₂ , 10% O ₂ , 85% Ar), 0 °C, 5 mg cat., 50 mL autoclave	528 (373)	12878	3.6 MPa (5% H ₂ , 95% Ar) / 3.6 MPa N ₂ , 0 °C, 5 mg cat.	0	0	Our work
Pd ₁ /PdSn-NW ^a	4.1 (0.9)	4 MPa (5% H ₂ , 10% O ₂ , 85% Ar), 0 °C, 5 mg cat., 60 mL autoclave	630	18529	3.6 MPa (5% H ₂ , 95% Ar) / 3.6 MPa N ₂ , 0 °C, 5 mg cat.	820	360	Our work
Pd ₆ Pb NRs/TiO ₂	3.1 (1.1)	4.0 MPa (5% H ₂ , 10% O ₂ , 45% Ar, 40% He), 0 ± 2 °C, 30 mg cat., 30 mL autoclave	(170.1)	5667	No test / 2 MPa (20% O ₂ , 80% He), 0 ± 2 °C, 30 mg cat.	~260	~0.9	ACS Catalysis 2021, 11 , 1106
AuPd@HZSM-5	2.4 (2.3)	40 bar (4% H ₂ , 20% O ₂ , 76% N ₂), 30 °C, ca. 32 mg cat., 300 mL autoclave	(320)	12800	29 bar (5% H ₂ , 95% CO ₂) / No test, 2 °C, 10 mg cat.	/	41	ACS Catalysis 2021, 11 , 1946
Pd ₃ Pb/s-TiO ₂	2.4 (1.7)	1 atm (2.5% H ₂ , 50% O ₂ , 47.5% Ar), 20 °C, 5 mg cat., 100 mL reactor	(176.1)	7339	/	/	/	ACS Catalysis 2021, 11 , 2288
1% Pd NP/C	1.3 (/)	4.0 MPa (5% H ₂ , 25% O ₂ , 70% CO ₂), 2 °C, 10 mg cat., 100 mL autoclave	(120)	12000	/	360	/	ACS Catalysis 2020, 10 , 5928
R-PdNi/TiO ₂ -C ^b	0.85 (0.42)	4.0 MPa (5% H ₂ , 25% O ₂ , 70% CO ₂), 2 °C, 10 mg cat., 100 mL autoclave	(96.6)	11364	/	/	/	ACS Catalysis 2021, 11 , 8407
3 wt % Pd-2 wt % Sn/TiO ₂	3 (2)	29 MPa (5% H ₂ , 95% CO ₂) / 1 atm (air), 2 °C, 10 mg cat.	(61)	2033	0	/	/	Science 2016, 351 , 965
2.5% Au-2.5% Pd/carbon	2.5 (2.5)	0	(160)	6400	0	0	0	Science 2009, 323 , 1037

^a Annealing temperature is 350 °C

^b The catalytic reaction solvent contains 50 mL D₂O and 0.5 M sulfuric acid

^c The parenthesis is the loading of the second metal in the Pd-M alloy catalysts

^d The parenthesis is the producibility of the catalysts performed in 30 min.

Revisions:

We have changed the Table 1 in the SI to the Table shown above.

Methods:

6) Line 392, page 23: Why should the autoclave used herein be a semibatch autoclave?

Reply: We agree with the reviewer that there is no difference between autoclave and semibatch autoclave in this work. We have therefore changed “semibatch autoclave” to “autoclave” in the Method section of the revised manuscript.

7) Line 411, page 14: Reaction arrow should be used here.

Reply: We thank the reviewer for pointing out this error. We have changed it to reaction arrow in the revised manuscript.

8) Line 412, page 14: How did the authors determine the H₂O concentration needed to calculate the H₂O₂ selectivity according to the equation reported in line 412?

Reply: Because H₂ was converted to H₂O₂ and H₂O in the reaction, we calculated the H₂O moles by subtracting the moles of H₂O₂ produced from the moles of H₂ converted. To address the

reviewer's concern, we have added a formula to explain how to calculate the H₂O moles in the revised manuscript.

Revisions:

In the line 466 of the Methods of the revised manuscript, we have added the following formula:

$$n(\text{H}_2\text{O})_{\text{out}} = n(\text{H}_2)_{\text{converted}} - n(\text{H}_2\text{O}_2)_{\text{out}}$$

9) The standard deviations of the measurements are impressively small (Figure 1). Does the standard deviation here refer to several repetitions of the same catalytic test? How often did the authors repeat each catalytic test?

Reply: Yes. The standard deviation here refers to the several repetitions of the catalytic test using fresh catalyst each time. The catalysts were repeated for 3-5 times to obtain the standard deviation.

Revisions:

In the line 143 of the caption of Fig. 1 of the revised manuscript, we have added “The standard deviation was achieved from the repeated runs of 3-5 times using fresh catalyst for each test.”

10) What is the metal loading of the TiO₂-supported catalysts? Figure 5 in the SI: “...Loadings of different Pd samples measured by ICP-OES analysis. ...” To which of the Pd samples do the authors refer here (i.e. the TiO₂-supported or unsupported worm structures)? Please, specify.

Reply: The Pd loading of the supported Pd_L/PdSn-NW catalyst is 4.1 wt.% and we have added the Pd loading into the revised Table 1 of the SI. We apologize that the reviewer did not see the sample names in Figure 5 of the SI, which were displayed inside the figure. To address the reviewer's comment, we have added a statement in the caption of the Figure 5 of the SI to specify it.

Revisions:

We have added the Pd loading of the catalysts into the revised Table 1 in the Supplementary materials.

We have added “The sample names are displayed in the Figure.” in the caption of the Figure 5 of the SI.

11) Did the authors perform control experiments to examine the possibility of interphase mass transfer constraints? If not, please state that explicitly.

Reply: Yes. We have performed control experiments to exclude the possibility of interphase mass transfer constraints. We changed the stirring rate (RPM) and found that there is no any change for the H₂O₂ productivity (figure below). Therefore, there is no interphase mass transfer constraints.

Supplementary Figure for the reviewers The comparison of H₂O₂ producibility of the Pd_L/PdSn-NW catalyst in the direct H₂O₂ synthesis under different stirring rate indicating there is no interphase mass transfer constraints under the reaction conditions reported.

Revisions:

We have added the above figure into the revised Supplementary Materials.

In the revised Method section, we have also added “Control experiments were performed by varying the stirring rate from 100-1200 rpm and it is confirmed that there is no interphase mass transfer constraints in the reaction under the present conditions (Supplementary Figure 26).”

I was wondering if the authors also tested the very nice nanowires that they synthesized by a similar procedure in Ref. 32 and if a similar effect of structural defects is also observed for direct H₂O₂ synthesis?

Reply: According to the reviewer’s suggestion, we have tested the very nice nanowires synthesized by a similar procedure in Ref. 32. As can be seen, the nice nanowires showed higher decomposition/decomposition activity. The catalytic data has been added in the revised Supplementary information.

Supplementary Figure for the reviewers TEM images of the nice PdSn nanowires and the comparison of H₂O₂ producibility of the nice PdSn nanowires and the Pd_L/PdSn-NW catalyst supported on TiO₂ in the direct H₂O₂ synthesis from O₂ and H₂.

Revisions:

We have added the above figure into the revised Supplementary materials.

In the line 125 of the revised manuscript, we have added “and a nice PdSn nanowires prepared by the approach reported in the literature”

In the line 126 of the revised manuscript, we have added “and low decomposition/hydrogenation activity (Supplementary Figure 10)”.

While this work shows some interesting results, significant revisions are required before publication in Nature Communications.

Reviewer #3 (Remarks to the Author):

Li et al. reports PdSn nanowire catalysts for efficient H₂O₂ direct synthesis where the nanowire is layered with Pd oxides. The nanowire catalysts provide superior reactivity in the H₂O₂ direct synthesis at zero Celcius, in particular, H₂O₂ selectivity of >95%. They claim that the superior

catalytic performance mainly results from the Pd oxides layered on PdSn nanowires. Although the topic is timely and interesting, I have several critical concerns which prevent the publication of this work. My comments are found below.

1. The role of the Pd oxides for H₂O₂ direct synthesis is clear. In particular, in DFT results (Figure 4), I don't make sure that the PdSn nanowires with Pd oxide layers provides the high H₂O₂ selectivity of >95%. In Figure 4d, the authors considered only the main reaction for H₂O₂ direct synthesis without considering side reactions (*OH formation, *O₂ dissociation, and dissociation of the formed H₂O₂). Without comparing the side reaction pathways, one cannot make sure the high H₂O₂ selectivity. And, in Figure 4d, the *HOOH formation on PdO@Pd₄Sn is thermodynamically exothermic by ~1.25 eV, which indicates that the detachment of the formed H₂O₂ is likely a rate-determining step for H₂O₂ direct synthesis on PdO@Pd₄Sn. In this work, the catalytic performance was evaluated at zero Celcius. Thus, I don't think that the 1.25 eV can be overcome at zero Celcius, which revealing a disagreement between the DFT calculations and experiments.

Reply: We thank the reviewer for the comments and apologize for the confusion about the desorption energy of *HOOH. It must be clarified that the absolute value of 1.25 eV is the reaction energy of the entire reaction process but not the desorption energy of *HOOH, and the desorption of H₂O₂ on PdO@Pd₄Sn requires an energy of only 0.63 eV (0.14 eV for free energy in Figure S32) as shown in Figure 4b. To address the reviewer's comments, we have performed additional simulations and discussion on the reaction pathways. We have performed the calculations on the side reactions (*OH formation, *O₂ dissociation, and dissociation of the formed H₂O₂ on Pd₄Sn (Figure S31) and PdO@Pd₄Sn (Figure S32)). In addition, all the DFT calculated energies are corrected to Gibbs free energies which include the entropy and zero-point energy (Figures shown below and in Supplementary materials). The computational details and the corresponding figures have been updated in the revised manuscript and SI. All the conclusions in the original submission are not changed in the revised manuscript.

Supplementary Figure for the reviewers (a) DFT optimized structures of PdO(101), Pd₄Sn and PdO@Pd₄Sn, (b) adsorption energies of H₂, O₂, H₂ + O₂, and H₂O₂, (c) free energy profiles for H₂ activation, (d) free energy profiles for O₂ reduction by the surface hydrides on these three models. TS: transition state. Color code: Pd, blue; O, red; H, white; Sn, yellow.

The calculation results showed that the *O₂ and *OOH dissociations on the Pd₄Sn(111) surface are barrierless, while the *HOOH dissociation has a small barrier of 0.07 eV, as shown in the Figure below. Therefore, all the three side reactions are easier to take place than the reaction of H₂O₂ production (Figure above, d), suggesting the poor selectivity of H₂O₂ on Pd₄Sn(111).

Supplementary Figure for the reviewers Optimized structures of IS, TS and FS of *O₂, *OOH and *HOOH dissociation catalyzed by Pd₄Sn(111), with free energy barriers (G_{TS}), reaction free energies (G_{r}) and adsorption free energies (G_{ads}). Color code: blue, Pd; yellow, Sn; red, oxygen; white, H.

On the PdO@Pd₄Sn surface, the simulation results of the side reactions are shown in Figure below. Different from the case on Pd₄Sn, the energy barriers of *O₂, *OOH and *HOOH dissociation on the PdO@Pd₄Sn surface are 1.74, 0.53 and 0.32 eV, respectively, which are much larger than those (0.53, 0.27 and 0.14 eV for O₂ hydrogenation, OOH hydrogenation and H₂O₂ desorption, respectively) of the corresponding competitive reactions (Figure below). It therefore suggests that the H₂O₂ selectivity on PdO@Pd₄Sn is high, which is consistent with the experimental observations.

Once again, we thank the reviewer for this important suggestion, which has improved this work.

Supplementary Figure for the reviewers Optimized structures of IS, TS and FS of *O₂, *OOH and *HOOH dissociation catalyzed by PdO@Pd₄Sn, with free energy barriers (G_{TS}), reaction free energies (G_{r}) and adsorption free energies (G_{ads}).

Revisions:

We have added the above figures in the revised manuscript and Supplementary materials.

In the line 308 of the revised manuscript, we have added “In addition, we simulated three main side reactions of *O₂, *OOH and *H₂O₂ dissociation on Pd₄Sn and PdO@PdSn and the corresponding structures of IS, TS and FS are shown in Supplementary Figures 31-32. On Pd₄Sn(111), the *O₂ and *OOH dissociations are barrierless while the *HOOH dissociation has a small barrier of 0.07 eV (Supplementary Figure 31), suggesting the poor selectivity of H₂O₂ on Pd₄Sn(111). On PdO@Pd₄Sn, the energy barriers of the three side reactions are 1.74, 0.53 and 0.32 eV (Supplementary Figure 32), respectively, which are larger than those of the corresponding competitive reactions (0.53, 0.27 and 0.14 eV for O₂ hydrogenation, OOH hydrogenation and H₂O₂ desorption, respectively, Supplementary Figures 30 and 32). Therefore, these results suggest the high selectivity of H₂O₂ on PdO@Pd₄Sn, which is consistent with the experimental observations (Fig. 1d and Table 1).”

In the line 499 of the revised manuscript, we have added the computational details for free energy calculations. We have added “The Gibbs free energy (G) was obtained by VASPKIT code via the following formula:

$$G = E + E_{\text{ZPE}} + \Delta U_{0 \rightarrow T} - T \times S$$

where E is the electronic energy of DFT calculations, E_{ZPE} and S are the zero-point energy and entropy computed from vibrational frequency analysis. T is the temperature adopted in our experiment (273.15 K). $\Delta U_{0 \rightarrow T}$ is the internal energy difference between 0 and T K.”

2. Although the authors performed XPS analyses and Bader charge calculations, the origin of Pd oxide layers for the superior catalytic performance is not still clear. The authors need to perform in-depth electronic structure analyses with DFT methods and then compare with XPS analyses.

Reply: We thank the reviewer for the suggestion. According to the reviewer’s comments, we have performed in-depth electronic structure analysis with DFT methods and compared with XPS analysis. Using DFT calculations, we have found that different interactions between catalyst and intermediates lead to different reactivity and selectivity. Because the surface Pd atoms on PdO(101) and PdO@Pd₄Sn have similar coordination environments, we therefore compared their electronic structures in detail. From Bader charge analysis results, one can see that the surface Pd on PdO(101) loses 0.67 electrons, while on PdO@Pd₄Sn the Pd is less positive, losing only 0.46 electrons, in agreement with the experimental trend observed (Fig. 2d).

Supplementary Figure for the reviewers Charge state distribution on PdO(101) and PdO@Pd₄Sn with atomic coloring according to their Bader charges.

To gain in-depth insight into the relationship between electronic structure and catalytic behavior, we further compared the projected density of states (PDOSs) of the surface Pd- d_{z^2} states on PdO(101) and PdO@Pd₄Sn since the d_{z^2} states are well recognized to play an important role in molecular adsorption (*ACS Catal.* 2020, 10, 7, 4377–4384; *Phys. Rev. B* 2007, 76, 235433; *Phys. Rev. B* 2001, 64, 085412). As can be seen in the Figure below, the distribution of d_{z^2} of surface Pd_{3c} on PdO is more localized than that of surface Pd_{2c} on PdO@Pd₄Sn. Furthermore, a sharp peak of d_{z^2} states exists across the Fermi level on PdO(101), implying a strong interaction between the surface Pd and the reaction species via donating/accepting electrons. In contrast, on PdO@Pd₄Sn, the distribution of Pd- d_{z^2} state is less significant around the Fermi level, indicating that the weaker binding of reaction species to Pd.

Supplementary Figure for the reviewers Projected density of state (PDOSs) of surface $\text{Pd}_{3c}\text{-}d_{z^2}$ on PdO and $\text{Pd}_{2c}\text{-}d_{z^2}$ on PdO@Pd₄Sn. Fermi level was set to 0 eV.

Revisions:

We have added the above figure about DOSs in the revised Supplementary materials.

In the line 273 of the revised manuscript, we have added “To gain in-depth insight into the link between electronic structure and catalytic behavior, the projected density of states (PDOSs) of the surface Pd d_{z^2} states on PdO(101) and PdO@Pd₄Sn were calculated and compared since the d_{z^2} states are well recognized to play an important role in molecule adsorption. As can be seen in Supplementary Figure 23, the distribution of d_{z^2} of surface Pd_{3c} on PdO is more localized than that of surface Pd_{2c} on PdO@Pd₄Sn. Furthermore, a sharp peak of d_{z^2} states exists across the Fermi level on PdO(101), implying the strong interaction between surface Pd and the reaction species via donating/accepting electrons. In contrast, on PdO@Pd₄Sn, the distribution of Pd- d_{z^2} state is less significant around the Fermi level, indicating that the binding of Pd to reaction species is weaker.”

3. In the Introduction section, the authors explained that nanowire structure would be able to provide a unique platform for catalysis. Because of this reason, the authors considered PdSn nanowire structures. However, in this work, the superior catalytic performance mainly results from Pd oxides layered on PdSn nanowires, not the nanowire structures themselves.

Reply: We agree with the reviewer that in this work, the superior catalytic performance mainly results from Pd oxides layered on PdSn nanowires, not the nanowire structures themselves. However, the PdSn nanowire plays as a template to generate layered Pd oxide, which is different from Pd oxides supported on other carriers. Therefore, the nanowire structure is beneficial to the formation of the Pd oxide layers and can be considered as a platform.

4. The authors evaluated catalytic performance of the nanowires for H₂O₂ direct synthesis at zero Celsius. Why did they consider zero Celsius? Usually, in the field of the catalysis for H₂O₂ direct synthesis, a room temperature has been considered. The authors need to evaluate catalytic

properties of the samples at room temperatures and then compare with reported other state-of-the-art catalysts.

Reply: We considered zero Celsius in the reaction because low temperature is safer and easier to carry out in the setup of the lab. There are many literatures focused on low temperature synthesis. For example, in Edwards' work (*Science* 2009, 323, 1037), the reaction temperature is 2 °C. The reaction temperature in Z. Jin's work (*ACS Catal.* 2021, 11, 1946) is 2 ± 2 °C and the temperature in Freakley's work (*Science* 2016, 351, 965) is 2 °C. However, to address the reviewer's comment, we have also evaluated the catalytic properties of the supported Pd_L/PdSn-NW catalyst at room temperature (Figure shown below) and compared with the other catalysts reported. These results have been included in the revised Table S1 in the revised Supplementary materials.

Supplementary Figure for the reviewers The H₂O₂ producibility comparison of the Pd_L/PdSn-NW catalyst carried out at 0 Celsius and 25 Celsius.

Revisions:

In the line 444 of the method section of the revised manuscript, we have added “The reaction was carried out at zero Celsius because it is much safer and easier to carry out in our lab. We also investigated the reactivity of the catalyst at room temperature for comparison. The producibility of the Pd_L/PdSn-NW catalyst at room temperature is slight lower than that performed at zero Celsius in the direct H₂O₂ synthesis (Supplementary Figure 33).”

We have also added the above figure into the revised Supplementary Materials.

5. The authors need to show the data regarding stability of the nanowire catalyst such as recyclability or/and long-term stability.

Reply: We have investigated the recyclability and long-term stability of the supported Pd_L/PdSn-NW catalyst and these data are shown below. The supported Pd_L/PdSn-NW catalyst is very stable in multiple runs and no deactivation was observed. The slight variation of the activity is caused by

the slight difference in the catalyst mass for each run (Table below). The long-term H₂O₂ producibility of the catalyst has also been added in the revised Supplementary materials.

Supplementary Figure for the reviewers The recyclability of the Pd_I/PdSn-NW catalyst in the direct synthesis of H₂O₂. C/C₀ is the ratio of H₂O₂ producibility produced in each run to that of the first run in the reaction cycles. The spent catalyst after each run was centrifuged and washed using ethanol/acetone mixture. After drying at 60 °C, the material was used for the next-cycle test. During the recycle experiments, there is the catalyst loss in each run and the catalyst mass of each cycle in the experiments of recycling is shown in the table below:

Catalyst	The catalyst mass of each cycle (mg)				
	1	2	3	4	5
Pd _I /PdSn-NW	5.07	4.84	4.58	4.27	3.99
PdSn-NP	5.02	4.87	4.64	4.32	4.09

Supplementary Figure for the reviewers The function of the H₂O₂ producibility with the time over the Pd_I/PdSn-NW catalyst in direct H₂O₂ synthesis from H₂ and O₂ at zero Celsius.

Revisions:

In the line 126 of page 4 of the revised manuscript, we have added “Furthermore, the annealed Pd₁/PdSn-NW catalyst is very stable in the reaction and no deactivation is found in multiple runs (Supplementary Figure 9).”

In the revised Supplementary materials, we have added the recyclability figure and Table shown above.

<Minor comments>

1. In DFT calculations, why did consider the PdO(101) or Pd₄Sn(111) surfaces? Are the surfaces thermodynamically most stable? And, the authors need to clarify the crystal structures of PdO and Pd₄Sn for DFT calculations.

Reply: Yes, the PdO(101) and Pd₄Sn(111) surfaces were considered because they are thermodynamically stable. For PdSn nanowires, since Pd(111) was considered as the most stable facet of Pd crystal in literatures (*React Kinet Catal Lett* 14, 61–65 (1980)) and an intense diffraction peak of Pd(111) against other facets was also found in Fig. 2a of the manuscript, Pd₄Sn (111) facet was chosen to simulate the Pd₄Sn slab. For PdO facet, previous experiments reported that the oxidation treatment of metal Pd led to the preferential growth of high quality PdO(101) film on Pd(111) (*Surface Science*, 2009, 603, 2671; *Surface Science*, 2008, 602, L53; *Chem. Rev.*, 2013, 113, 4164). It suggests that the lattice of PdO(101) and Pd(111) matches very well. Therefore, the model of Pd₄Sn(111) covered with PdO(101) was chosen in this work.

According to the reviewer’s comment, we have also clarified the crystal structures of PdO and Pd₄Sn for DFT calculations in the revised manuscript. To simulate PdO, an optimized PdO unit cell ($a = b = 3.056 \text{ \AA}$; $c = 5.381 \text{ \AA}$, in agreement with experimental value: 3.043 and 5.335 \AA) was adopted to cleave (101) facets (Figure below, a and b). (2×3) slab containing 4 O-Pd-O tri-atomic layers with 15 \AA vacuum space was adopted to simulate the PdO(101) slab. Since the Pd unit is composed of 4 Pd atoms, uniform Pd_xSn_y can only be constructed as Pd₂Sn₂ and Pd₃Sn. In addition, our experiment shows that nanowires expose more Pd on the surface, we therefore simulate the surface with 4 layers of uniform Pd₃Sn covered with a layer of Pd (Figure below, c and d). Then, the total Pd/Sn ratio equals to 4:1 and maintains a uniform distribution of Pd and Sn ions in the xy plane. The optimized lattice parameter ($a=b=c$) for Pd₃Sn is 4.047 \AA , which is consistent with that in the Materials Project database (<https://materialsproject.org/materials/mp-718/>).

Supplementary Figure for the reviewers Optimized models of the catalysts used in the DFT simulations: (a) PdO unit cell, (b) PdO(101), (c) Pd₃Sn and (d) Pd₄Sn (111). Color code: red, O; blue, Pd; yellow, Sn.

Revisions:

In the line 489 of the computational details of the revised manuscript, we have updated the description as “The (2×3)-PdO(101) surface was cleaved from the optimized PdO unitcell ($\underline{a} = \underline{b} = 3.056 \text{ \AA}$, $\underline{c} = 5.381 \text{ \AA}$, in agreement with experimental values ($\underline{a} = \underline{b} = 3.043 \text{ \AA}$ and $\underline{c} = 5.335 \text{ \AA}$)), containing 4 O-Pd-O tri-atomic layers. The bottom two layers were fixed while the other layers with surface species were fully relaxed in structural optimization. The lengths of a and b of supercell are 12.38 and 9.17 \AA , respectively. Since the Pd unit is composed of 4 Pd atoms, uniform Pd _{x} Sn _{y} can only be constructed as Pd₂Sn₂ and Pd₃Sn. In addition, our experiments show that nanowires expose more Pd on the surface, we therefore simulate the slab with 4 layers of uniform Pd₃Sn covered with one layer of Pd (Supplementary Figure 31). Then, the total Pd/Sn ratio equals 4:1. For all slabs, a vacuum space of 15 \AA is adopted along the z -direction to avoid the interaction between periodic images.”

2. On line 52, the "PtSn" nanocatalyst should be changed into "PdSn".

Reply: We thank the reviewer for pointing out this typo. We have changed “PtSn” to “PdSn” in the revised manuscript.

REVIEWER COMMENTS

Reviewer #1 (Remarks to the Author):

I thank the authors for considering the comments made in the previous assessment of this manuscript – and for completing additional experiments. The new data demonstrate some limitations of the catalytic system.

If a H₂O₂ synthesis catalyst is stable in terms of activity and selectivity the productivity measured should be constant over time – the new results reported at different reaction times show that the productivity is decreasing over time – which suggests that either the catalyst is i) deactivating ii) as the H₂O₂ concentration is increasing the so is the rate of the subsequent degradation reactions or iii) the reaction is becoming H₂ limited. This suggests that the high initial activity and selectivity could be of kinetic origin (low H₂O₂ concentration leading to low degradation reaction rates) rather than a truly selective catalyst. This effect also seems to be temperature sensitive. Can these catalysts achieve high rates and selective H₂O₂ efficiency in the presence of high concentrations of H₂O₂ or at least constant productivity as H₂O₂ is accumulated – the new data might suggest not? In which case these materials just represent similar effects to those reported by Hutchings et al rather than a step change.

In the stability testing I assume C/C₀ should be a percentage? Or equal to 1 or below?

The red circles included in the new figure S2 are not clear what they represent – how can the authors tell the PdSn is in these areas?

The EXAFS is still unclear – the XPS shows that the majority of the Sn is present as SnO_x and as the nanowires are 10 nm – this could be considered bulk as close to bulk analysis – but the EXAFS is fitted to Pd-Pd and Pd-Sn scattering features – which structural models were used for this modelling? Why were Pd-O and Sn-O not considered? There still seems inconsistency between the XPS and EXAFS.

Reviewer #2 (Remarks to the Author):

Haifeng Xiong and coworker have thoroughly revised their manuscript. I have a few comments on the revised manuscript:

1) Table 1 in the supporting information:

- The same units (e.g. with respect to reaction pressure etc.) should be used throughout the table.
- It's not completely clear which conditions refer to which type of catalyst (please, change the format slightly).

2) To demonstrate absence of gas-liquid mass transfer, the authors carried out tests at varying stirring speed. For the three phase system, I suggest that they additionally carry out corresponding tests with varying catalyst mass.

Reviewer #3 (Remarks to the Author):

The revision appropriately reflects responses to all of my criticisms. Therefore, I can recommend this revision for the publication of Nature Communications.

Response to the Reviewers' comments

GENERAL: We thank the reviewers for all their comments, which help us to improve the manuscript. We have considered all of them carefully and have revised the manuscript accordingly. The changes are colored in red in the manuscript. The detailed point-to-point responses are listed below.

REVIEWER COMMENTS

Reviewer #1 (Remarks to the Author):

I thank the authors for considering the comments made in the previous assessment of this manuscript – and for completing additional experiments. The new data demonstrate some limitations of the catalytic system.

If a H₂O₂ synthesis catalyst is stable in terms of activity and selectivity the productivity measured should be constant over time – the new results reported at different reaction times show that the productivity is decreasing over time – which suggests that either the catalyst is i) deactivating ii) as the H₂O₂ concentration is increasing the so is the rate of the subsequent degradation reactions or iii) the reaction is becoming H₂ limited. This suggests that the high initial activity and selectivity could be of kinetic origin (low H₂O₂ concentration leading to low degradation reaction rates) rather than a truly selective catalyst. This effect also seems to be temperature sensitive. Can these catalysts achieve high rates and selective H₂O₂ efficiency in the presence of high concentrations of H₂O₂ or at least constant productivity as H₂O₂ is accumulated – the new data might suggest not? In which case these materials just represent similar effects to those reported by Hutchings et al rather than a step change.

Reply: We agree with the reviewer that there is the observed decrease over time for the H₂O₂ productivity (rate). The recycling experiments (Supplementary Figure 9) show that the Pd_L/PdSn-NW catalyst is very stable under the reaction conditions. Therefore, the slowdown in the H₂O₂ productivity with time is not due to the catalyst deactivation. For this reaction, the H₂O₂ productivity (reaction rate, r) can be expressed as: $r = k \cdot (P_{H_2})^a (P_{O_2})^b (H_2O_2)^c$, where k is the rate constant, P_{H₂} and P_{O₂} are the partial pressures of H₂ and O₂, H₂O₂ is H₂O₂ concentration. The H₂O₂ rate is very related to both the H₂/O₂ partial pressures and the H₂O₂ concentration. Therefore, the observed decrease in reaction rate is due to the decrease of the reactant gases (H₂/O₂) and the increase of H₂O₂ concentration in the autoclave over time, which is determined by the reaction equilibrium, regardless of the catalyst. The effect of reactant gas pressure on the reaction rate is also confirmed by the extra experiments we performed below.

To address the reviewer's comments, we have performed extra experiments by varying the experiment procedures and autoclave volumes. We have firstly performed the reaction for 15 min. Then, the reactant gas in the reactor was degassed and re-pressurized using H₂/O₂ to run the reaction

for the second 15 min to achieve the H₂O₂ productivity in 30 min. The degassing and re-pressurizing were repeatedly used to achieve the H₂O₂ productivity in 45 min and 60 min, respectively. The data of the H₂O₂ productivity using this approach are shown in the figure a below (blue bars). As can be seen, using the above approach, there is no decrease in the H₂O₂ productivity between 15 min and 30 min runs. The H₂O₂ productivity only shows slight decrease after 30 min, as compared to the sharp decrease using the conventional method by directly testing the reaction for a certain time (figure b below). Therefore, the assumption of the H₂/O₂ in the autoclave over time is the one of the reasons of the decreased H₂O₂ productivity observed in the system. It is also confirmed by the tests that the H₂O₂ productivity in 60 min using a 100 mL autoclave (orange bar in figure a below) is higher than that obtained in 45 min and 60 min using a 50 mL autoclave (blue bar in figure a below) because the former has more H₂/O₂ molecules. Furthermore, the slight decrease of H₂O₂ productivity after 30 min in figure a below is ascribed to the accumulation of H₂O₂ in the autoclave. Therefore, the reaction rate cannot be compared under different reactant concentrations and different times in an autoclave reactor. However, the data reported in the work clearly showed that the Pd_L/PdSn-NW catalyst is highly selective without the side reactions (hydrogenation and decomposition) and present the highest rate in the H₂O₂ direct synthesis under the same conditions, as compared to the literature.

More importantly, different from the 3-dimensional PdO particles reported by Hutchings et al., this work reports that a layer of metal oxide (Pd oxide layer, 2-dimensional PdO) located on nanowires (Pd_L/PdSn-NW), displaying superior reactivity in the H₂O₂ direct synthesis, which has the potent to extend to other catalytic reactions.

Supplementary Figure for the reviewers The comparison of H₂O₂ producibility of the Pd_L/PdSn-NW catalyst in the direct H₂O₂ synthesis over time. (a) the tests were performed by repeatedly degassing and re-pressurizing at each 15 min. The data with blue color was obtained from a 50 mL autoclave and the data with orange color was obtained from a 100 mL autoclave. (b) The tests were performed by extending the reaction time in a 50 mL autoclave. The catalyst mass used in these reactions are 5 mg.

Revisions:

To address the reviewer's comments, in the revised Supplementary materials, we have included the above figures and added the explanations of "We have performed extra experiments by varying the experiment procedures and autoclave volumes. We have firstly performed the

reaction for 15 min. Then, the reactant gas in the reactor was degassed and re-pressurized using H_2/O_2 to run the reaction for the second 15 min to achieve the H_2O_2 productivity in 30 min. The degassing and re-pressurizing were repeatedly used to achieve the H_2O_2 productivity in 45 min and 60 min, respectively. The data of the H_2O_2 productivity using this approach are shown in Supplementary Figure 11a (blue bars). As can be seen, using the above approach, there is no decrease in the H_2O_2 productivity between 15 min and 30 min runs. The H_2O_2 productivity only shows slight decrease after 30 min, as compared to the sharp decrease using the conventional method by directly testing the reaction for a certain time (Supplementary Figure 11b). Therefore, the assumption of the H_2/O_2 in the autoclave over time is the one of the reasons of the decreased H_2O_2 productivity observed in the system. It is also confirmed by the tests that the H_2O_2 productivity in 60 min using a 100 mL autoclave (orange bar in Supplementary Figure 11a) is higher than that obtained in 45 min and 60 min using a 50 mL autoclave (blue bar in Supplementary Figure 11a) because the former has more H_2/O_2 molecules. Furthermore, the slight decrease of H_2O_2 productivity after 30 min in Supplementary Figure 11a is ascribed to the accumulation of H_2O_2 in the autoclave. However, these two issues (H_2/O_2 moles and H_2O_2 concentration in the solvent) are related to the reaction equilibrium, instead of the catalyst deactivation (Supplementary Figure 9) or catalyst property.”

In the revised manuscript, we have added “It should be mentioned that the H_2O_2 productivity slightly slows down over time and it is ascribed to the decrease of the reactant gas concentration and the accumulation of H_2O_2 in the autoclave (Supplementary Figure 11), which is not due to the catalyst deactivation.”

In the stability testing I assume C/C_0 should be a percentage? Or equal to 1 or below?

Reply: We thank the reviewer for pointing out this issue. Yes, it should be percentage. We have changed it in the revised Supplementary Figure 9.

The red circles included in the new figure S2 are not clear what they represent – how can the authors tell the PdSn is in these areas?

Reply: We agree with the reviewer that the bright field TEM images are not clearly shown the PdSn on TiO_2 . Therefore, we have performed dark-field STEM images to show the morphology of the PdSn on TiO_2 , as shown below.

Supplementary Figure for the reviewers STEM images of Pd_I/PdSn-NW catalyst.

Revisions:

In the revised Supplementary Figure 2, we have added the above STEM images.

The EXAFS is still unclear – the XPS shows that the majority of the Sn is present as SnO_x and as the nanowires are 10 nm – this could be considered bulk as close to bulk analysis – but the EXAFS is fitted to Pd-Pd and Pd-Sn scattering features – which structural models were used for this modelling? Why were Pd-O and Sn-O not considered? There still seems inconsistency between the XPS and EXAFS.

Reply: We apologize for the confusion. Yes, the XPS shows that the majority of the Sn is present as SnO_x. However, we considered the XPS as a surface-sensitive analysis because the XRD results clearly indicated that the bulk phase of the PdSn nanowire after annealing in air is metallic Pd (Fig. 2a) and neither bulk SnO_x nor bulk PdO was formed. Therefore, the EXAFS is fitted to Pd-Pd and Pd-Sn scattering features because both XRD and XAS are bulk techniques. Moreover, HRTEM image (Supplementary Figure 14a) shows that only a layer of PdO was present and the lattice fringe of the bulk phase is ascribed to the PdSn alloy. Therefore, Pd-O and Sn-O were not considered. To address the reviewer's concern, we have added some statement in the revised manuscript to explain this issue.

Revisions:

In the revised manuscript, we have added “The EXAFS is fitted to Pd-Pd and Pd-Sn scattering features because the XRD results clearly indicate that the bulk phase of the PdSn nanowire after annealing in air is metal Pd (Fig. 2a) and both XRD and XAS are bulk techniques. Furthermore, the HRTEM image shows the lattice fringes corresponding to metallic PdSn nanowire (Supplementary Figure 14a).”

Reviewer #2 (Remarks to the Author):

Haifeng Xiong and coworker have thoroughly revised their manuscript. I have a few comments on the revised manuscript:

1) Table 1 in the supporting information:

- The same units (e.g. with respect to reaction pressure etc.) should be used throughout the table.

Reply: We thank the reviewer for the comment. We have used the same units in the revised Table 1 in the supporting information and have changed “bar” to “MPa” for the reaction pressure unit.

- It's not completely clear which conditions refer to which type of catalyst (please, change the format slightly).

Reply: According to the reviewer's comment, we have changed the format slightly so that the conditions corresponding to the type of catalyst is clear now.

2) To demonstrate absence of gas-liquid mass transfer, the authors carried out tests at varying stirring speed. For the three phase system, I suggest that they additionally carry out corresponding tests with varying catalyst mass.

Reply: According to the reviewer's suggestion, we have additionally carried out corresponding tests with varying catalyst mass. We have performed the reaction with the catalyst mass from 3-7 mg in a 50 mL autoclave and 10 mg catalyst in a 100 mL autoclave. These new results are displayed below. As can be seen, the H₂O₂ productivity did not change with varying catalyst mass, indicating there is no gas-liquid mass transfer.

Supplementary Figure for the reviewers The comparison of H₂O₂ productivity of the Pd₁/PdSn-NW catalyst in the direct H₂O₂ synthesis with varying catalyst mass indicating there is no interphase mass transfer constraints under the reaction conditions reported. The data with the light blue were carried out in a 50 mL autoclave and the data with the red color was obtained from a 100 mL autoclave.

Revisions:

In the revised Supplementary Figure 35, we have added the above figure to further demonstrate the absence of gas-liquid mass transfer.

In the revised manuscript, we have changed “Control experiments are also performed by varying the stirring rate from 100-1200 rpm. It is confirmed that there are no interphase mass transfer constraints in the reaction under the present conditions (Supplementary Figure 35).” to “Control experiments are also performed by varying the stirring rate from 100-1200 rpm, varying the catalyst mass from 3-10 mg using autoclaves having different volumes (50 mL and 100 mL). All these experiments show the similar H₂O₂ producibility under these conditions (Supplementary Figure 35). Therefore, it indicates that there are no interphase mass transfer constraints in the reaction under the present conditions.

Reviewer #3 (Remarks to the Author):

The revision appropriately reflects responses to all of my criticisms. Therefore, I can recommend this revision for the publication of Nature Communications.

Reply: We thank the reviewer for approving our revisions.

REVIEWER COMMENTS

Reviewer #1 (Remarks to the Author):

The authors performed additional experiments to reveal the nature of the catalyst activity decreases - by re-charging the autoclave at 15 min intervals the effect of gas depletion was eliminated - however after 30 min the productivity decreases by 20%. This suggests catalyst deactivation or H₂O₂ degradation - which should be fully acknowledged by the authors in the manuscript in a better way than ". Furthermore, the slight decrease of H₂O₂ productivity after 30 min in Supplementary Figure 11a is ascribed to the accumulation of H₂O₂ in the autoclave."

According to their kinetic expressions they consider H₂O₂ to be a poison to the catalyst with a negative exponent - this may not be the case and in fact the catalyst just begins to decompose H₂O₂ as the concentration increases. So either the catalyst changes in the presence of H₂O₂ and either becomes less active or begins to degrade H₂O₂.

Reviewer #2 (Remarks to the Author):

The authors have addressed all my concerns and edited the manuscript accordingly.

Response to the Reviewers' comments

REVIEWER COMMENTS

Reviewer #1 (Remarks to the Author):

The authors performed additional experiments to reveal the nature of the catalyst activity decreases - by re-charging the autoclave at 15 min intervals the effect of gas depletion was eliminated - however after 30 min the productivity decreases by 20%. This suggests catalyst deactivation or H₂O₂ degradation - which should be fully acknowledged by the authors in the manuscript in a better way than ". Furthermore, the slight decrease of H₂O₂ productivity after 30 min in Supplementary Figure 11a is ascribed to the accumulation of H₂O₂ in the autoclave."

Reply: We agree with the reviewer that the H₂O₂ productivity (rate) decreased after 30 min, which was ascribed to the accumulation of H₂O₂ in the autoclave. According to the reviewer's suggestion, we have changed ". Furthermore, the decrease of H₂O₂ productivity after 30 min in Supplementary Figure 11a is ascribed to the accumulation of H₂O₂ in the autoclave." to ". Furthermore, the decrease of H₂O₂ productivity (rate) after 30 min in Supplementary Figure 11a possibly resulted from the accumulation of H₂O₂ in the autoclave according to Le Chatelier's principle or the decomposition of H₂O₂ or catalyst deactivation at high H₂O₂ concentrations." in the page 13 of the revised Supplementary Materials.

According to their kinetic expressions they consider H₂O₂ to be a poison to the catalyst with a negative exponent - this may not be the case and in fact the catalyst just begins to decompose H₂O₂ as the concentration increases. So either the catalyst changes in the presence of H₂O₂ and either becomes less active or begins to degrade H₂O₂.

Reply: We thank the reviewer for the comments. According to the reviewer's suggestion, we have made the change as described in the reply above. We have also made slight change in the manuscript to point out this issue mentioned by the reviewer.

Revisions:

In the line 135 of the page 4 of the revised manuscript, we have changed ", which is not due to the catalyst deactivation." to ". Besides, it possibly suggests either the catalyst changes in the presence of high concentration H₂O₂ and either becomes less active or begins to degrade H₂O₂ at high H₂O₂ concentrations."

Reviewer #2 (Remarks to the Author):

The authors have addressed all my concerns and edited the manuscript accordingly.

Reply: We thank the reviewer for approving our revisions.

REVIEWERS' COMMENTS

Reviewer #1 (Remarks to the Author):

The authors have now included the statements i made about the paper in the paper however .